# Discovery of F-18 labeled repurposed CNS drugs by computational strategy for effective tau imaging and alzheimer's diagnosis

Pan Tang[1], Xuehua Chen[2], Pingping Li[3], Ling Zhang[4], Bin Tang[4], Jun Wen[1], Yan Liu[1], Iqra Kalsoom[5], Chong Cheng [2]*

1 Department of Neurology,Changde Hospital, Xiangya School of Medicine, Central South University (The First People's Hospital of Changde), Changde, China, 2 Department of Nuclear Medicine, Changde Hospital, Xiangya School of Medicine, Central South University (The First People's Hospital of Changde), Changde, China, 3 Department of Paediatrics, Changde Hospital, Xiangya School of Medicine, Central South University (The First People's Hospital of Changde), Changde, China, 4 Centre for Reproductive Medicine, Changde Hospital, Xiangya School of Medicine, Central South University (The First People's Hospital of Changde), Changde, China, 5 School of Chemistry and Chemical Engineering, Beijing Institute of Technology, Beijing, China

* j001cg@163.com

## Abstract

Alzheimer's disease (AD) remains a significant challenge in diagnosis and treatment, with current methods insufficient for early detection. A major obstacle is the lack of effective imaging agents targeting the Tau protein, which plays a key role in AD pathology. To address this, we developed a computational methodology for selecting F-18 labeled drug candidates from a library of CNS-penetrant compounds curated from literature and databases. The library, consisting of 977 compounds, was evaluated based on clinical data, target proteins, pathways, toxicity, and other relevant factors. We implemented Python-based algorithms to select the top 39 compounds from virtual screening results, prioritizing those with optimal Tau binding affinity and BBB permeability. Additionally, we developed an algorithm to identify F-18 labeling candidates that maintain their biological activity post-labeling. We then performed virtual screening of an F-18 labeled drug library and selected the top 3 compounds based on stability and imaging potential. The selected compounds exhibited molecular weights of 350–520 Da, favorable logP values (2.05–2.72), and high BBB permeability. Our findings indicate that Drug 388 (BI-D1870), binds to Tau with a binding free energy of ΔG = −8.79 kcal/mol. Drug 416 (reported BRAF inhibitor, RG6344) shows a binding free energy of ΔG = −7.91 kcal/mol, while Drug 610 (Iloperidone/ HP 873), a D2/5-HT2 receptor antagonist, exhibits a predicted binding free energy of ΔG = −6.88 kcal/mol with the target Tau protein respectively. Molecular dynamics simulations confirmed stable Tau-drugs interactions, with minimal RMSD fluctuations, indicating strong binding. The F-18 label enabled real-time PET imaging, allowing non-invasive tracking of the drug's binding to Tau in the brain. Our approach provides

**Data availability statement:** All relevant data are within the paper and its Supporting Information files.

**Funding:** This work was supported by an active grant from the Changde Science and Technology Bureau (Technology R&D and Innovation Guidance Program; Grant/Award Number: Changke Han [2022] No. 51) with project title "Application of Modified Volume-Viscosity Swallow Test in Neurodegenerative Diseases". The funders had no role in study design, data collection and analysis, decision to publish, or preparation of the manuscript.

**Competing interests:** The authors have declared that no competing interests exist.

a comprehensive solution to the current limitations in Alzheimer's diagnosis by offering F-18 labeled drugs that effectively target Tau protein without compromising their biological activity, advancing both diagnostic and therapeutic strategies for AD.

## Introduction

Alzheimer's disease (AD) stands as the predominant form of dementia, representing a global health challenge with significant clinical and societal impact [1]. It is a progressive neurodegenerative disorder that impacts cognition function as well as behaviour [2]. According to the World Health Organization (WHO), over 55 million people worldwide suffer from AD or other types of dementia, and every year there are nearly 10 million new cases [3]. In the United States, about one out of nine individuals (10.8%) age 65 and older have AD, with occurrence of 1275 new cases per 100,000 person annually [4]. There is yet no cure for AD unfortunately and patients are usually diagnosed at a late stage with an average survival period of 4–8 years [1,5].

Pathologically, Alzheimer's disease is a progressive disorder characterized by the deposition of extracellular plaques composed of aggregated forms of the amyloid-beta (Aβ) polypeptide and intraneuronal neurofibrillary tangles (NFTs) composed of aggregated hyperphosphorylated tau protein [6]. Tau pathology is marked to correlate more with neuronal loss, neurodegeneration and disease progression, making it a critical target for disease staging and therapeutic monitoring [7]. As Early and accurate diagnosis in AD remains a challenge, Nuclear medicine offers a powerful tool offering valuable insights into disease progression and diagnosis [8,9]. Several radiopharmaceuticals are employed in advanced imaging techniques, such as positron emission tomography (PET) and single-photon emission computed tomography (SPECT), to enable non-invasive and real-time visualization of the molecular processes [10–12].

Nuclear medicine is experiencing a radical shift with the emergence of theragnostic by integrating diagnostic imaging with targeted therapy [13,14]. Several Aβ- and tau-targeting radiotracers such as [$^{18}$F]florbetapir (Amyvid®), [$^{18}$F]florbetaben (Neuraceq®), [$^{18}$F]flutemetamol (Vizamyl®), and [$^{18}$F]T807 (Tauvid®) have received FDA approval for PET imaging in patients with cognitive impairment [15,16]. Imaging biomarkers have demonstrated clinical utility but also have some certain limitations [17]. Some tracers have shown off target binding to monoamine oxidase enzymes and melanin containing region, limited CNS penetration, slow elimination rates, nonspecific uptakes that reduces diagnostic clarity [18].

In CNS drug discovery, the assessment of brain exposure is critical for lead optimization as the compounds need to cross the blood-brain barrier (BBB) to reach their therapeutic targets [19]. The BBB is a complex system involving passive and active mechanisms of transport and efflux transporters [20,21]. Several PET tracers have been developed that are tau specific but most of them have certain limitations like poor CNS penetration, nonspecific binding, inadequate selectivity for pathological

tau over other proteins [22]. The heterogeneity and complex structure of tau aggregates in AD necessitates highly specific and bioavailable compounds with well-defined pharmacokinetic profile [23]. Hence, using CNS penetrant compound library and evaluating the physicochemical properties like molecular weights,we aim to evaluate existing shortcomings in this regard [24]. In addition to it, the development of a florine-18 labelled compound from CNS penetrant chemical library offers target specificity, high positron yield, favourable half-life and compatibility with radiochemistry [25].

In this context, we are persuading to extract a library of CNS penetrant compounds that could be radio labelled using Florine-18 radionucleotide [26]. In silico studies for the potential candidates from library and their virtual screening are performed, allowing efficient prioritization of molecules with better blood brain bioavailability [27–29]. In addition Molecular modelling and molecular docking studies were performed with Subsequent ADMET predictions that provide pharmacokinetic profiles, including blood-brain barrier permeability, metabolic stability, and safety parameters [30]. Our findings demonstrate that F-18 labelled drugs, 388 (BI-D1870) binds with Tau protein with binding free energies of $\Delta G = -8.79$ kcal/mol, drug 416 BRAF inhibitor/RG6344 have binding free energy of $\Delta G = -7.91$ kcal/mol and the 610 (Iloperidone/HP 873) having the predicted binding free energy of $\Delta G = -6.88$ kcal/mol respectively [31,32]. By integrating CNS drug-likeness parameters with radiolabeling feasibility, this study seeks to prioritize compounds with high translational potential for development as diagnostic agents in nuclear medicine, thereby accelerating the preclinical screening process and supporting the advancement of novel imaging probes for Alzheimer's disease.

## Materials and methods

This study employed an integrated computational pipeline to identify and optimize CNS-penetrant compounds for tau-targeted AD diagnostics and therapeutics. Virtual screening of a curated library of blood-brain barrier (BBB)-permeable drugs (SDF format) was conducted using MOE (Molecular Operating Environment), with molecular docking against the tau protein. Compounds were preprocessed for physiological pH (7.4) and evaluated for drug-likeness (LogP: 2.5–3.8; MW: 350–520 Da). Top hits were assessed for 18F-radiolabeling feasibility via a custom RDKit script, which identified fluorine sites for isotopic substitution. ADMET properties (e.g., BBB penetration, solubility) were predicted using MOE and SwissADME. Finally, molecular dynamics (MD) simulations (100 ns, Amber99sb force field) analyzed binding stability for prioritized 18F-labeled candidates. This workflow synergized structure-based screening, radiolabeling prediction, and dynamic validation to advance tau-directed PET probes and therapeutics.

### Retrieving BBB permeable drugs library and virtual screening against tau protein

To construct an initial virtual screening library for blood-brain barrier (BBB)-penetrant drugs targeting Tau protein, we conducted a structured literature mining using PubMed. The search employed key terms such as "blood-brain barrier crossing drugs", "clinically tested drugs", and "drugs with known targets and pathways". From the results, 977 compounds were selected based on established or predicted ability to cross the BBB, documented clinical development status, and well-characterized molecular targets and pathways. The compounds were curated to ensure relevance in neurological applications and potential imaging adaptability, particularly for PET. Each selected drug was annotated with information including molecular weight, SMILES structure, target classes (e.g., mTOR, CDK, autophagy modulators), biological activity, and clinical phase status. The library was designed with the intent to identify 18F-labelable candidates for subsequent PET radiotracer development (S1 Fig). Moreover, a virtual screening approach was implemented to identify potential candidates for diagnostic imaging and therapeutic treatment of AD. The screening was performed on curated library of CNS-penetrant drugs, using MOE (Molecular Operating Environment) for molecular docking. The drug database included substances with recognized blood-brain barrier (BBB) penetration properties and was supplied in SDF format. MOE's structure preparations tools were used to create and validate the SDF file that contained a library of CNS medications. To make that the right ionization modes were allocated for docking simulations, the compounds were hydrogenated at physiological pH (7.4). After obtaining the Tau protein structure via the Protein Data Bank PDB) (PDB ID: 9EOH) [33],

 

MOE's energy-minimizing methodology was used to refine the structure and prepare it for docking. By removing any steric conflicts or unfavorable bond geometries from the structure, this step guaranteed the stability of the target protein. The molecular weights ranged from 350 to 520 Da, that is the optimal range for compounds that resemble drugs. Values for solubility fell within acceptable bounds for intravenous delivery. All of the chosen drugs showed appropriate LogP values (between 2 and 4), confirming high blood-brain barrier (BBB) penetration.

### Radiolabeling prediction for F-18 labeled compounds

The next step after the virtual screening was to determine whether fluorine-18 (F-18) labeling for imaging using PET was feasible. Since F-18 is frequently utilized in PET scans because of its advantageous qualities, the F-18 labeling procedure was carried out by substituting Fluorine (F) atoms in the discovered high-affinity molecules with Fluorine-18 (F-18). To find the functional groups in the evaluated compounds that included fluorine, a custom Python script was created using RDKit. This script evaluated each chemical for fluorine (F) atoms and projected the possibility of labeling with fluorine-18 by switching the F atoms with F-18. The substances with the most effective labeling positions were identified as candidates for PET imaging. This script evaluated the SDF database of top docking hits, produced new SDF files with the labeled compounds, and exposed them to a further round of virtual screening.

### Molecular docking studies

Molecular docking has become an important method in the recognition and creation of lead compounds, offering a quick and economical alternative to laboratory analysis. Virtual screening, which examines the binding interactions and processes of multiple compounds, is critical to this technology [34–36]. The desired Tau protein and compound library was docked employing the MOE docking module. The atoms of hydrogen were incorporated to the target protein structure, and energy minimization was performed using MOPAC7.0. The ligands were docked in flexible mode, whereas the receptor (Tau) remained rigid. Docking calculations were run to determine each compound's binding potential using MOE's affinity scoring tool. The docking data were ordered by binding energy, and the leading candidates were chosen for their favourable binding interactions with the Tau binding site.

### ADMET and drug-likeness prediction of the target F-18 labeled drugs

Drug-likeness screening is required to identify potential therapeutic compounds in the early stages of drug development. Identifying the chemicals' ADMET (Absorption, Distribution, Metabolism, Excretion, and Toxicity) characteristics is a key step in the process. To determine the efficacy of these F-18 tagged compounds for in vivo application, ADMET parameters were anticipated. The properties of logP (lipophilicity), the molecular weight, the solubility, and BBB permeability were estimated utilizing MOE and RDKit. The above characteristics were utilized to determine that the selected drugs had good CNS penetration and could be useful choices for PET imaging and therapy. For additional confirmation of the validity of pharmacological ADMET analysis, researchers can use the simple SwissADME platform (http://www.swissadme.ch/) to assess the possible physiological consequences of therapeutic pharmaceuticals.

### Molecular dynamics simulations for tau binding

The binding strength and dynamics of the leading three F-18 labeled candidate drugs, including 388 (BI-D1870), drug 416 (BRAF inhibitor/RG6344), and drug 610 (Iloperidone/HP 873), were evaluated using molecular dynamics (MD) models. The simulation process was conducted using MOE with the Amber99 force field [37,38]. MD simulations were run for a period of 100 ns and we performed triplicate MD simulations for each drug to evaluate the stability of the drug-Tau complex over time. The top three F-18 labeled compounds were first docked into the Tau binding pocket and the resulting protein-ligand complex was solvated using the TIP3P water model. Counterions were added to neutralize the system.

Energy minimization was performed to remove any steric clashes, followed by NVT (constant volume and temperature) and NPT (constant pressure and temperature) equilibration. The MD simulations were run at 300 K temperature and 1 bar pressure for 100 ns. The systems were monitored for RMSD (Root Mean Square Deviation) analysis to evaluate the stability of both the ligand and protein complexes during the simulation. To validate the reproducibility of RMSD data from MD simulations across three drugs (318, 416, and 610), statistical analyses were performed on combined data from three independent simulation replicas per drug. The simulations for each drug spanned 0–20,000 configurations, with measurements every 10 units, resulting in approximately 2,200–2,215 data points per replica. The datasets included the Hamiltonian (H), representing the total system energy measured during MD using MOE 2009–2010, along with potential energy (U), kinetic energy (K), and RMSD values for the ligand, binding pocket (or alpha carbon for drug 318), and alpha carbon (AC). For drug 610, the solvent-accessible surface area (ASA) was also available but excluded from primary analyses. Descriptive statistics, such as mean, standard deviation (SD), and coefficient of variation (CV = SD/mean × 100%), were calculated for H and RMSD metrics at each time point, across entire simulations, and collectively across all drugs to assess central tendency and variability. Convergence was evaluated by computing running averages and SDs over a 100-time-point moving window to confirm stabilization. Inter-replica variability was quantified through pairwise differences and SDs between replicas per drug, with one-way ANOVA applied to test for significant differences in means ($p > 0.05$ indicating reproducibility) both within each drug and across datasets. Pearson correlation coefficients were computed for pairwise comparisons of H and RMSD trends across replicas and drugs to evaluate consistency. Time-averaged values were derived from the equilibrated phase (last 10% of each simulation) and compared using t-tests or ANOVA to confirm similarity across replicas and drugs. RMSD trends across drugs were compared by downsampling replica-averaged values to 1,000 ps intervals for visualization and further analysis. Significant ANOVA results ($p < 0.05$) prompted post-hoc Tukey's Honestly Significant Difference (HSD) tests to identify pairwise differences between drugs, using all equilibrated phase data points ($n \approx 660$ per drug) and controlling the family-wise error rate at $\alpha = 0.05$. All analyses were performed using Python libraries such as NumPy, SciPy, Pandas, and Statsmodels.

## Results and discussion

The final retrieved and assembled library comprised 977 BBB-penetrant drug candidates with diverse mechanisms of action relevant to neurodegeneration and Tau pathology, including kinase inhibitors (e.g., CDK and mTOR inhibitors), autophagy regulators, and apoptosis inducers [39]. Notable examples include Flavopiridol (Phase 2, CDK inhibitor), PQR620 (mTOR inhibitor), and SNS-032 (CDK inhibitor, Phase 1). Prior evidence also indicates that CDK5 inhibition can reduce neurotoxicity in AD models [40–42]. Each compound was assessed for molecular weight, biological targets, and clinical phase data, alongside SMILES representations to facilitate in silico screening. Based on their pharmacological profile, known safety data, and chemical structures amenable to F-18 substitution, these compounds serve as a rationally designed set for docking studies and radiolabeling potential. The next step involved computational binding affinity evaluation with Tau protein, followed by selection of the top three candidates for F-18 labeling and PET imaging feasibility studies. Ensuring BBB permeability was essential for translational feasibility, as this property directly impacts the success of CNS-targeted radiopharmaceuticals [43]. Additionally, selecting molecules with F-18 labeling potential aligns with current trends in developing dual-purpose PET tracers that serve both diagnostic and therapeutic roles.

### Virtual screening results for Tau

After conducting the virtual screening of 977 compounds using MOE, 39 compounds from the CNS-penetrant drug library were identified as strong binders to Tau (Fig 1). These compounds exhibited binding affinities with Tau ranging from −6.00 to −9.0 kcal/mol, demonstrating favorable interactions with the target protein. Specifically, the top 10 drug candidates had hydrophobic interactions and hydrogen bonds with the Tau microtubule-binding region, which is crucial for maintaining Tau's function and stability. These 39 compounds were successfully modeled with $^{18}$F atoms incorporated at viable positions, preserving

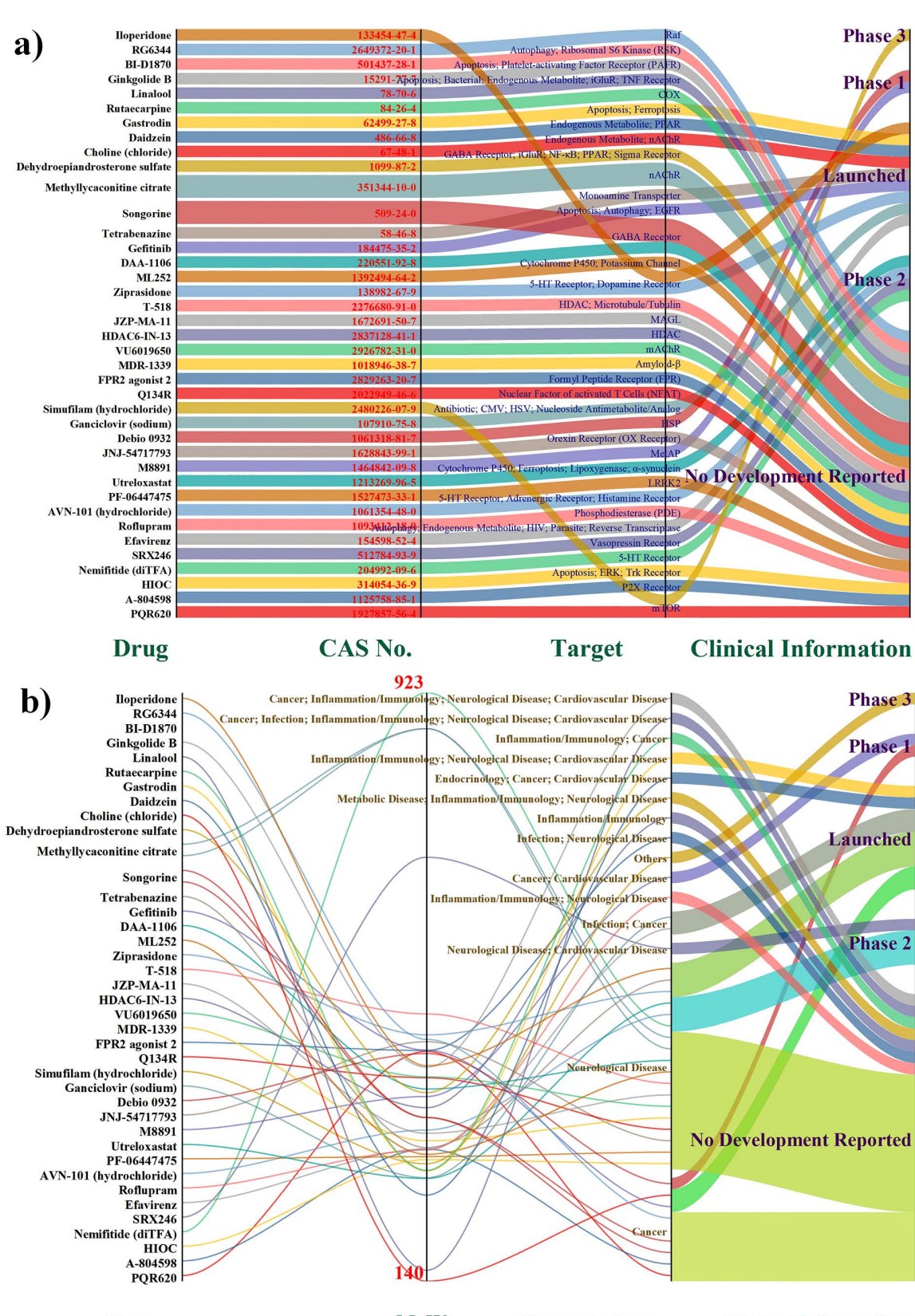

**Fig 1. Visualization of the top 39 drugs selected after the first phase of virtual screening, based on binding free energies with tau, ADMET (Absorption, Distribution, Metabolism, Excretion, Toxicity), p-values, biological activities, and blood-brain barrier (BBB) permeabilities. (a)** A Sankey diagram representing the progression of the selected drugs through different clinical phases (Phase 1, Phase 2, Phase 3, and launched). Each drug is shown with its respective CAS number and target, with the flow indicating its development status and phase. **(b)** A summary diagram highlighting the key characteristics of the selected drugs, including their molecular weights, research areas, clinical information, and the clinical development phases they are in.

chemical stability and pharmacophore integrity. Subsequent ADMET profiling further refined the selection of three compounds with molecular weights between 425–459 Da, within the acceptable drug-like range (350–520 Da), and demonstrated favorable solubility for intravenous delivery. Most importantly, logP values ranged from 2.05 to 2.72, indicating high BBB permeability, a critical parameter for neuroimaging agents. As visualized in S1 Fig a (clinical phase vs. research area) and e/f (molecular weight vs. clinical phase), these three candidates also aligned well with existing drug development trends and CNS-penetrant physicochemical space. Together, their strong Tau affinity, labeling feasibility, favorable ADMET properties, and CNS compatibility support their prioritization for future PET probe development in neurodegenerative disease research.

The virtual screening results demonstrate the robustness of the computational workflow in identifying potential Tau-binding ligands with docking energies comparable to clinically established tracers such as [¹⁸F]MK-6240 and [¹⁸F]PI-2620 [44,45]. The presence of hydrophobic and hydrogen-bond interactions within the Tau microtubule-binding domain supports the likelihood of selective binding, a critical feature for imaging specificity. Additionally, the favorable logP range (2.0–3.0) and preserved pharmacophore integrity following F-18 modeling suggest that the top-ranked compounds possess the optimal balance of BBB permeability, solubility, and structural stability necessary for CNS imaging applications. These results collectively indicate that the identified candidates are promising scaffolds for developing next-generation Tau PET tracers with improved sensitivity and clinical translatability.

## Current status of selected drugs

The selected drugs, 388 (BI-D1870), 416 (RG6344), and 610 (Iloperidone) presented in Fig 2, have shown promising potential in crossing the blood-brain barrier (BBB), making them significant candidates in neurological and cancer

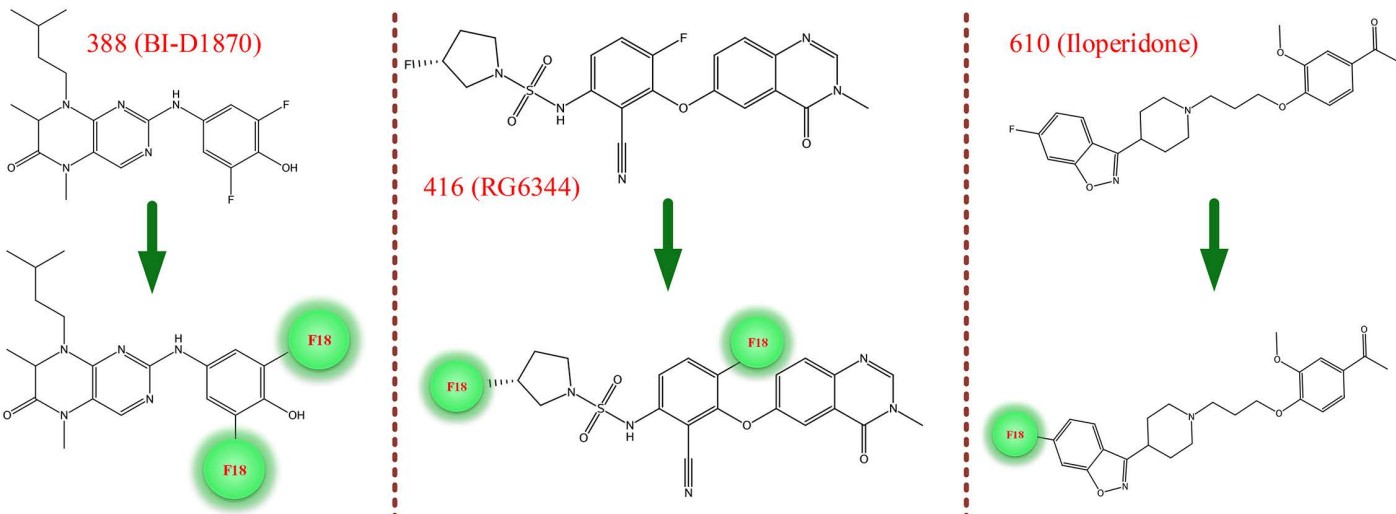

**Fig 2. Structural representations of the top three compounds selected in the virtual screening process, shown both with and without the F-18 isotope label.** The compounds 388 (BI-D1870), 416 (RG6344), and 610 (Iloperidone) are depicted in their standard forms (top) and with the F-18 isotope (bottom), highlighted in green circles.

research. BI-D1870, an ATP-competitive inhibitor of ribosomal S6 kinase isoforms, has demonstrated potent activity with low nanomolar IC50 values for all RSK isoforms, indicating its potential as a therapeutic agent for autophagy and MAPK/ERK pathway-related disorders [46]. However, there is currently no development report available for this drug. RG6344, a potent BRAF inhibitor, is specifically designed for targeting BRAF V600-mutant solid tumors, such as colorectal cancer [47]. It is known for its activity in the MAPK/ERK pathway, but like BI-D1870, no clinical development has been reported so far. Iloperidone, an atypical antipsychotic drug, is already launched and used for the treatment of schizophrenia symptoms. It acts as a dual antagonist for both dopamine and serotonin receptors, contributing to its efficacy in neurological disorders [48,49]. Despite their varied therapeutic targets, these drugs demonstrate effective blood-brain barrier penetration, which makes them valuable for further exploration in both cancer and neurological disease research. However, it is important to note that while Iloperidone has already been marketed, the other two drugs, BI-D1870 and RG6344, are still in the preclinical or early-stage development phase [50,51]. The strong BBB permeability of BI-D1870, RG6344, and Iloperidone supports their potential as CNS imaging probes. BI-D1870's modulation of the MAPK/ERK-RSK pathway is particularly relevant to Tau hyperphosphorylation and AD pathology. RG6344's BRAF inhibition may similarly influence Tau-related signaling, while Iloperidone's proven CNS safety profile enhances its translational viability. Collectively, these findings highlight their promise as repurposed candidates for Tau-targeted PET imaging.

## Interaction analysis of drugs with Tau protein

The F-18 labeled drug 388 (BI-D1870) interacts with several key amino acid residues of the Tau protein as shown in Fig 3a. The interaction diagram gives a detailed spatial view of how the drug interacts at the molecular level with the Tau protein structure, possibly inhibiting or modulating the Tau protein's role in Alzheimer's disease. The drug likely binds through hydrogen bond and hydrophobic interactions with the LEU-357, MET-337, PRO-332, GLN-336, GLU-338, LYS-340, GLY-355, LYS-331 and VAL-339, which prefer to interact with nonpolar groups crucial for the drug's action. The presence of alkyl or aromatic groups in the drug molecule likely facilitates these interactions that help stabilize the drug-protein complex. Further, the polar functional groups such as carbonyl or hydroxyl groups of the drug forms a hydrogen bond with LYS-331. The predicted binding free energy of the best three poses of drug-Tau complexes is ΔG=−8.79 kcal/mol, −8.10 kcal/mol and −7.85 kcal/mol respectively (S2 Fig). The drug is shown to bind effectively to these residues, forming critical interactions that is key to the drug's potential role in treating Alzheimer's disease, as it may disrupt Tau aggregation or improve brain function.

The Fig 3b illustrates the binding between an F-18 labeled drug 416 (BRAF inhibitor/RG6344) and the Tau protein. The diagram shows that the drug interacts with several specific amino acid residues on the Tau protein, including HIS-329, ASN-327, PRO-332, LYS-331, GLU-338 and LYS-340. These residues play a crucial role in the drug's effectiveness by forming various molecular interactions. Hydrophobic interactions are likely established between the drug and the nonpolar amino acid residues including PRO-332, LYS-331, LYS-340, ASN-327 and HIS-329, helping to anchor the drug within the hydrophobic pocket of the Tau protein. Moreover, the negatively charged GLU-338 residue may engage in electrostatic interactions with positively charged groups on the drug, further enhancing binding affinity. Van der Waals forces also contribute to the interaction, particularly with residues like PRO-332. The predicted binding free energy of the best three poses of drug-Tau complexes is ΔG=−7.91 kcal/mol, −7.73 kcal/mol and −7.25 kcal/mol respectively (S3 Fig). The F-18 isotope attached to the drug 416-BRAF inhibitor/RG6344 plays a significant role in its detection, enabling the drug's distribution and interaction with Tau to be monitored through PET scans.

The binding interaction between F-18 labeled drug 610 (Iloperidone/HP 873) and the Tau protein is shown in Fig 3c. The Drug 610 (Iloperidone/HP 873) interacts with key residues in the Tau protein, including HIS-329, GLU-338, LYS-340, and LYS-331, each contributing to the drug's binding stability. HIS-329 and LYS-331 forms both hydrogen bonds with the drug's polar functional groups, anchoring it to the protein. Additionally, the carboxyl group of GLU-338 participates in electrostatic interactions with the drug's positively charged regions, enhancing the binding affinity. These interactions are

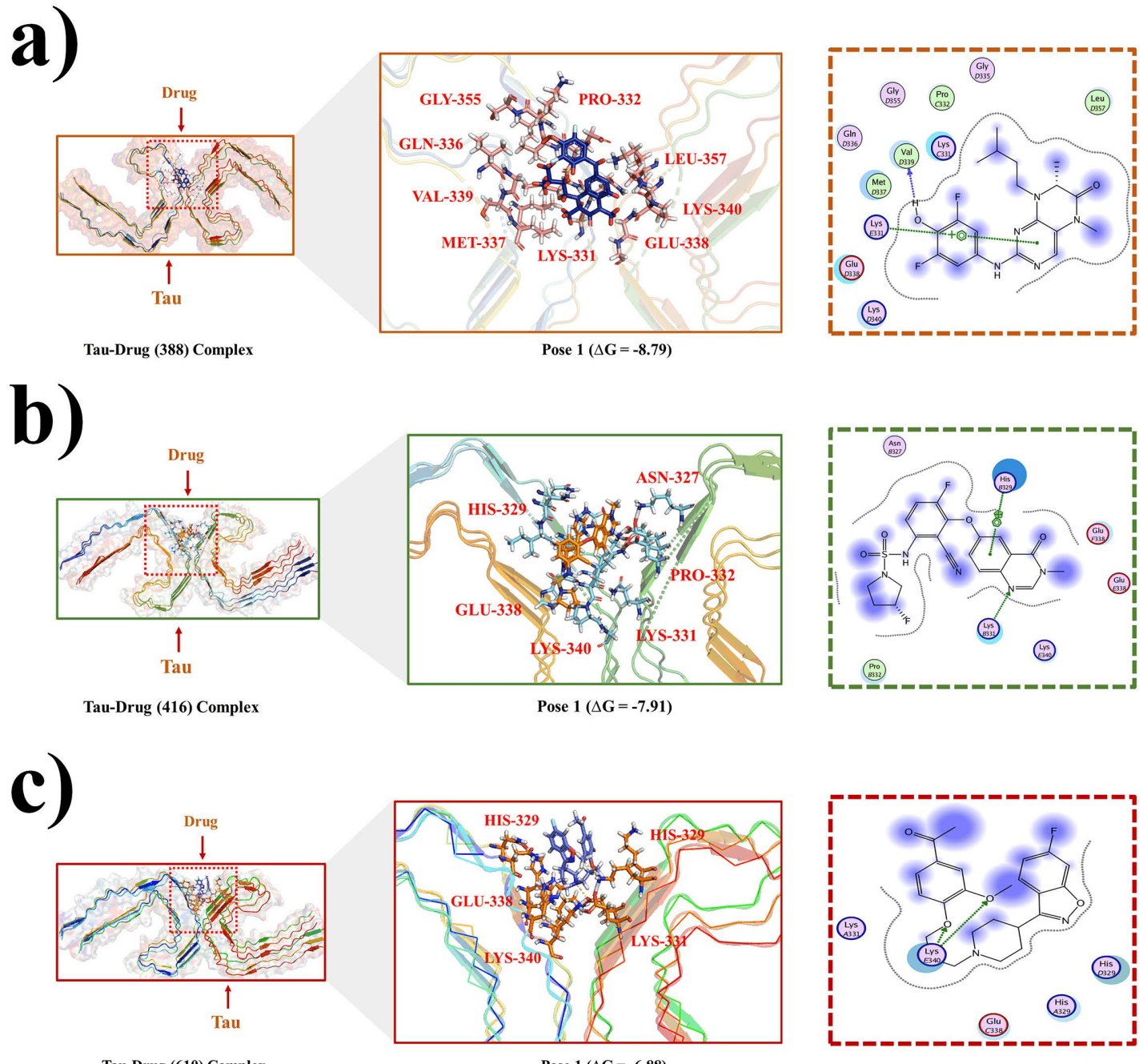

**Fig 3. Interaction analysis of the Tau protein with the 388 (BI-D1870), BRAF inhibitor/RG6344 and Drug 610 (Drug 610 (Iloperidone/HP 873)) drugs complexes. a)** 2D and 3D interaction diagrams of the docked complex of the drug 388 (BI-D1870) with Tau protein is presented. The left side displays closeup pictures of drug binding to Tau, proteins, highlighting hydrogen bonds and hydrophobic interactions. **b)** Drug (BRAF inhibitor/RG6344) interaction diagram in the active binding pocket of the Tau protein (both 2D and 3D). **c)** 2D and 3D interaction diagrams of Drug 610 (Drug 610 (Iloperidone/HP 873))-drug inside the Tau protein's active binding region.

crucial for the effective binding of the Drug 610 (Iloperidone/HP 873) to the Tau protein, suggesting its potential as a therapeutic agent for Alzheimer's disease. The predicted binding free energy of the best three poses of drug-Tau complexes is ΔG = −6.88 kcal/mol, −6.50 kcal/mol and −6.29 kcal/mol respectively (S4 Fig). The docking score of all the complexes with CBD is presented in Table 1.

The interaction analysis highlights the strong and specific binding of all three compounds BI-D1870, RG6344, and Iloperidone to key residues within Tau's microtubule-binding region. These residues (e.g., LYS-331, GLU-338, PRO-332) are critical for Tau aggregation, and their engagement suggests potential to interfere with fibril formation. The low binding free energies (−6.2 to −8.8 kcal/mol) indicate stable and high-affinity interactions comparable to reported Tau inhibitors [31,52]. Among them, BI-D1870 exhibited the strongest interaction network and stability, reinforcing its suitability as a Tau-targeted radioligand. Overall, these findings suggest that F-18 labeling preserves molecular integrity while enabling visualization of Tau interactions, supporting their potential use as PET imaging probes in Alzheimer's disease. Hence, the radiolabeled nature of the drug (F-18) enables its tracking through positron emission tomography (PET) imaging and monitoring the drug's distribution in the body, particularly in the brain, where Tau pathology is prominent. The radiolabeling with F-18 enhances the ability to visualize and track the drug's behavior in vivo, which is valuable for assessing its effectiveness in clinical settings. This combination of targeted drug binding and the diagnostic capability of F-18 labeling makes the drug an effective candidate for both treating Alzheimer's disease and monitoring treatment efficacy in clinical settings.

### MD simulation analysis

The docked complexes of the top three F-18 labeled drugs (388 (BI-D1870), BRAF inhibitor/RG6344, and Drug 610 (Iloperidone/HP 873)) with the Tau receptor underwent MD simulations to assess the stability of their protein-ligand complexes. The simulations were conducted for 100 ns, with the MD simulation being performed three times for each drug to ensure the reproducibility and reliability of the results. The RMSD analysis of the ligand, binding pocket, and protein in complex with the drug provides valuable insights into the structural stability and dynamics of the system throughout the simulation.

### RMSD analysis of Tau with drug 388 (BI-D1870) complex

The RMSD analysis of the Tau-388 (BI-D1870) complex shows that the ligand undergoes noticeable fluctuations in the first 0–10 ns (configuration 0–5000), indicating adjustments within the binding pocket (Fig 4a). From 10 to 30 ns (configuration 5000–10000), the RMSD stabilizes, with minor changes observed around 35 ns (configuration 15000). After 50 ns (configuration 20000), the ligand's RMSD remains consistent, confirming stable binding. The protein shows initial fluctuations between 0 and 20 ns (configuration 0–8000), stabilizing after 20 ns (configuration 8000), with slight deviations

**Table 1. Docking score and interacting binding residues of Tau protein with different labelled drugs.**

| Drugs | Receptor | Complex Poses | Binding Free Energy (Kcal/mol) | Amino Acid Residues involved in binding interactions |
|---|---|---|---|---|
| 388 (BI-D1870) | Tau | Pose 1<br>Pose 2<br>Pose 3 | −8.79<br>−8.10<br>−7.25 | LEU-357, MET-337, PRO-332, GLN-336, GLU-338, LYS-340, GLY-355, LYS-331 and VAL-339 |
| BRAF inhibitor/ RG6344 | – | Pose 1<br>Pose 2<br>Pose 3 | −7.91<br>−7.73<br>−7.85 | HIS-329, ASN-327, PRO-332, LYS-331, GLU-338 and LYS-340 |
| Drug 610 (Drug 610 (Iloperidone/HP 873) | – | Pose 1<br>Pose 2<br>Pose 3 | −6.88<br>−6.50<br>−6.29 | HIS-329, GLU-338, LYS-340, and LYS-331 |

*(Pose 2 and 3 of all drugs are presented in S2, S3 and S4)*

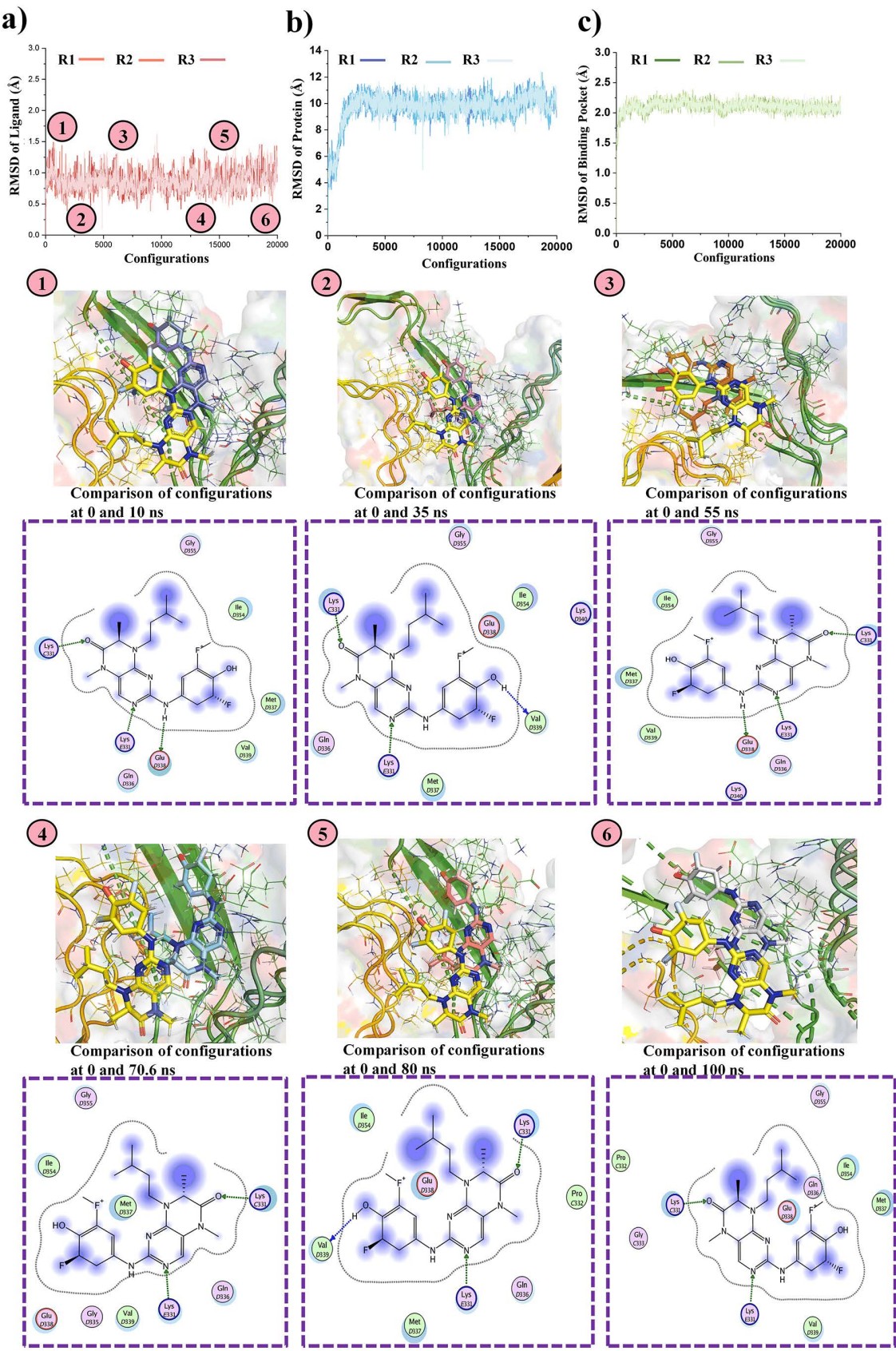

**Fig 4. RMSD analysis of the 388 (BI-D1870)/Tau protein complex during a 100 ns.** The integrity of the whole network is shown by **a)** the RMSD of drug (388 (BI-D1870)), **b)** the RMSD of the receptor protein, and **c)** the RMSD of the binding region. The small structural changes of the protein [1–6] demonstrate that binding of the target drug did not causes larger changes in the protein structure.

around 40 ns (configuration 16000). The protein structure becomes stable, reflecting the ligand's effect (Fig 4b). The binding pocket shows significant fluctuations between 0–10 ns (configuration 0–5000), stabilizing after 10 ns (configuration 5000), with minor fluctuations after 30 ns (configuration 12000), indicating that the pocket maintains a stable conformation (Fig 4c). The energy profile reveals that the total energy (H) decreases from 0 to 5000 configurations, stabilizing between −120 to −140 kcal/mol, potential energy (U) stabilizes around 500 kcal/mol, and kinetic energy (K) fluctuates around 5000 kcal/mol, showing minimal fluctuations and indicating a stable, energetically favorable protein-ligand complex (S5a, S5b and S5c Fig). Overall, the complex remains stable across all three replicates (R1, R2, and R3).

### RMSD analysis of Tau with drug 416 BRAF inhibitor/RG6344 complex

In the early stages of the simulation, the ligand's RMSD shows significant fluctuations between 0–5 ns (configuration 0–2500) as it adjusts its position within the binding pocket. From 5 to 25 ns (configuration 2500–12500), the RMSD decreases and stabilizes, indicating that the ligand has found a stable binding pose (Fig 5a). By 55 ns (configuration 22000), the ligand's RMSD remains steady, with minor fluctuations, suggesting it is firmly anchored. The F-18 label does not disturb the ligand's conformation, allowing for stable attachment and proper PET imaging. The protein's RMSD fluctuates slightly from 0 to 25 ns (configuration 0–12500) as it adjusts to the ligand binding. After 25 ns (configuration 12500), the protein stabilizes, with minor fluctuations and a slight increase at 40 ns (configuration 16000), indicating conformational adjustments (Fig 5c). The binding pocket shows fluctuations from 0 to 25 ns (configuration 0–12500), reflecting its flexibility, then stabilizes after 25 ns (configuration 12500), with minor fluctuations continuing after 55 ns (configuration 22000) (Fig 5b). The total energy (H), potential energy (U), and kinetic energy (K) remain consistent within a specified range, confirming the system's overall stability (S6a, S6b and S6c Fig).

### RMSD analysis of Tau with drug 610 (Iloperidone/HP 873) complex

The RMSD of the Tau/Drug 610 (Iloperidone/HP 873) complex shows significant fluctuations during 0–5 ns (configuration 0–2500) as the ligand adjusts within the binding pocket (Fig 6a). Between 5–25 ns (configuration 2500–12500), the RMSD stabilizes, indicating that the ligand has found a stable position, with minor fluctuations continuing until 55 ns (configuration 22000). This suggests the ligand is securely anchored and that the F-18 label does not affect its binding. The protein's RMSD fluctuates from 0 to 25 ns (configuration 0–12500) as it adapts to the ligand, stabilizing after 25 ns (configuration 12500), with minor deviations observed around 40 ns (configuration 16000). The protein reaches a stable conformation thereafter (Fig 6b). The binding pocket shows notable fluctuations between 0–25 ns (configuration 0–12500), stabilizing by 25 ns (configuration 12500) and continuing with minor fluctuations beyond 55 ns (configuration 22000), indicating retained flexibility (Fig 6c). Throughout the simulation, total energy, potential energy, and kinetic energy fluctuate within a narrow range, confirming the system's overall stability (S7 Fig).

 The MD simulations confirmed the structural stability and strong binding of all three F-18-labeled drug-Tau complexes, with minimal RMSD fluctuations and consistent energy profiles, indicating thermodynamic equilibrium throughout the 100 ns trajectory. Among the three, BI-D1870 exhibited the lowest RMSD values and the most stable protein-ligand complex, suggesting robust and persistent binding interactions. These results are consistent with previous studies demonstrating that lower RMSD and energy convergence correlate with high ligand-receptor stability in Tau-targeted systems [52,53]. The findings indicate that F-18 labeling does not destabilize ligand conformation or binding, supporting the feasibility of these compounds as reliable PET imaging probes for monitoring Tau aggregation in Alzheimer's disease.

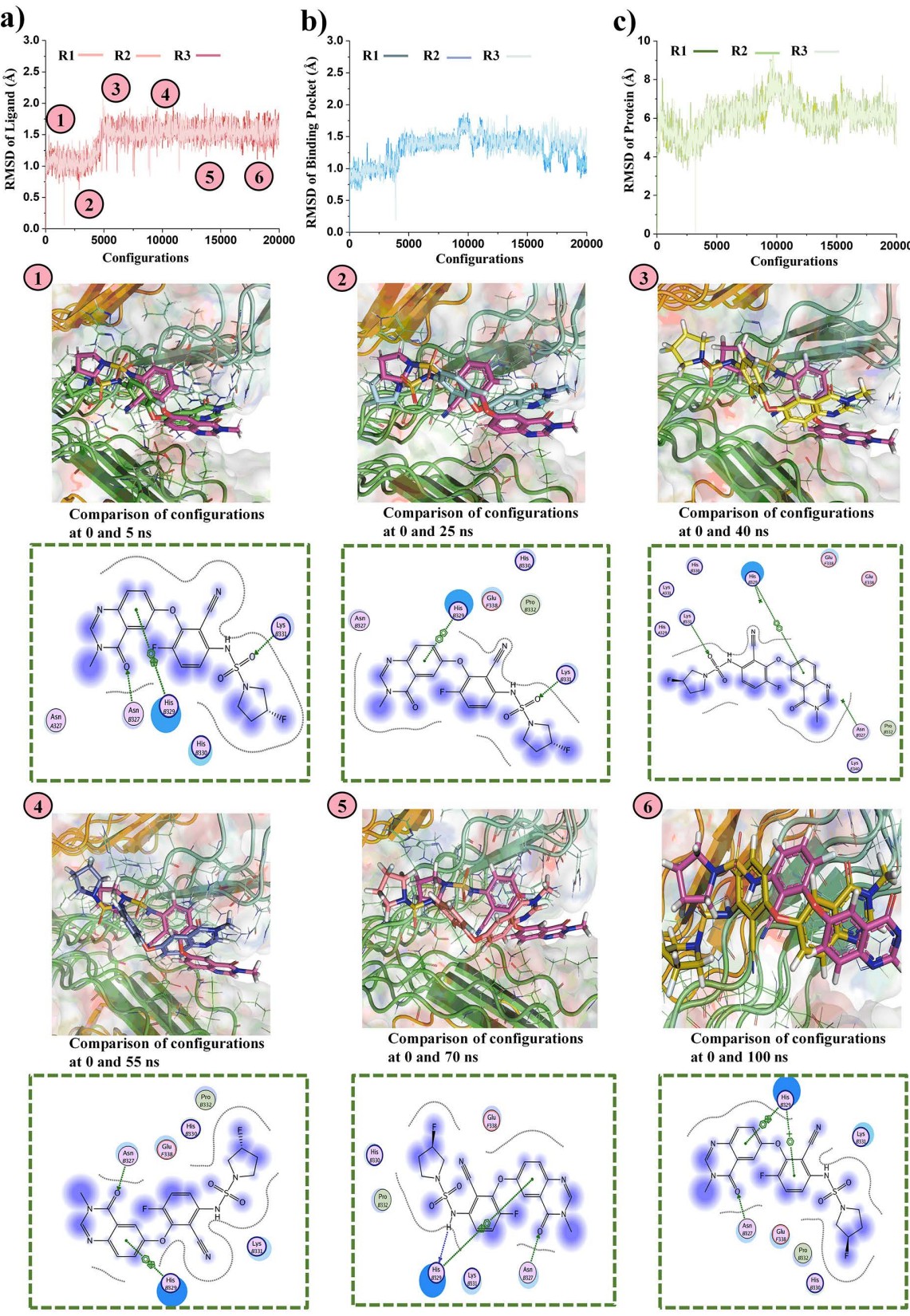

**Fig 5. RMSD studies and the drug's interaction of the Tau protein with the F-18 labeled drug (BRAF inhibitor/RG6344).** Key interactions between the target drug and the protein, including hydrogen bonding and hydrophobic interactions, are highlighted in the 2D interaction Figs. **a)** the RMSD of drug (BRAF inhibitor/RG6344), **b)** the RMSD of the receptor protein, and **c)** the RMSD of the binding region. The RMSD analysis demonstrates the strong binding, which is essential for therapeutic use in Alzheimer's disease.

The statistical analysis confirmed the high reproducibility of RMSD data across the three replicas for each drug, with low variability in the mean H values for drugs 318, 416, and 610. For the last 100 time points of each simulation, the mean H values were consistent, with drug 318 ranging from −103.33 to −104.89 kcal/mol, drug 416 from 173.0 to 173.5 kcal/mol, and drug 610 from 182.7 to 183.5 kcal/mol. RMSD values for the ligand ranged from 0.85 to 0.87 Å (CV 10–12%) for drug 318, 1.55 to 1.60 Å (CV 9–10%) for drug 416, and 2.90 to 3.00 Å (CV 6–7%) for drug 610 (S1 Table). RMSD of the pocket ranged from 2.0 to 2.1 Å (CV ~5%) for drug 318, 1.30 to 1.32 Å (CV 5–6%) for drug 416, and 2.50 to 2.55 Å (CV 3–4%) for drug 610 (Table 2, S2 Table). RMSD of the protein (Alpha carbon) was 10.0–10.8 Å (CV 5–7%) for drug 318, 6.0–6.2 Å (CV 6–7%) for drug 416, and 4.90–5.00 Å (CV 9–10%) for drug 610 (S3 Table). Convergence was achieved after ~20,000 configurations, with minimal inter-replica differences (|H| < 5 kcal/mol), and ANOVA confirmed no significant differences for H, RMSD ligand, and RMSD pocket (p > 0.05). Significant differences were observed for RMSD alpha carbon (p ≈ 0.03–0.04). Tukey's HSD test identified all pairwise differences as significant (p < 0.001), with drug 318 showing the lowest RMSD values across all metrics and drug 610 exhibiting the highest. Time-averaged RMSD values for the equilibrated phase were: ligand 318: 0.895 ± 0.143 Å, 416: 1.548 ± 0.122 Å, 610: 2.970 ± 0.153 Å; pocket 318: 2.094 ± 0.061 Å, 416: 1.189 ± 0.128 Å, 610: 2.582 ± 0.088 Å; Protein 318: 10.543 ± 0.569 Å, 416: 5.959 ± 0.528 Å, 610: 5.060 ± 0.464 Å (S4 Table).

These results affirm the reliability and robustness of the docking and MD simulations, with drug 318 showing the most stable ligand-protein interactions, drug 416 exhibiting intermediate stability, and drug 610 showing the greatest flexibility, particularly in the ligand and pocket. The statistical significance (p < 0.0001) and effect sizes (e.g., 2.076 Å ligand RMSD gap between 318 and 610) validate the differences in binding affinity and structural adaptability among the drugs, supporting their differential therapeutic potential.

## ADMET analysis of the top target F-18 labeled drugs

The Fluorine-18 labeling of the selected compounds was successful, as indicated by the radiolabeling prediction. The Fluorine-18 isotope was attached to the fluorine-containing functional groups of the drugs, making them suitable for PET imaging. A potent lead chemical that targets a particular receptor should be quickly eliminated from the body and exhibit adequate intestinal absorption. The idea of drug-likeness acts as a crucial foundation for drug discovery. In pharmaceutical studies, Lipinski's Rule of Five (RO5) is frequently employed to assess pharmacokinetic features and drug-likeness, focusing on oral bioavailability. A drug-like compound must have a molecular weight under 500, no more than five hydrogen bond donors, ten acceptors, a logP under 5, and only one violation [54–56]. In the present research, we used computational techniques to assess CBD's pharmacological characteristics, such as its ADMET profile. The LogP values range from 2.05 (BRAF inhibitor/RG6344) to 2.72 (Drug 610/Iloperidone), indicating ideal membrane permeability and good bioavailability. All drugs show positive water solubility, with Drug 610 having the highest (5.47), suggesting faster absorption. All three drugs exhibit high gastrointestinal absorption and are BBB permeant, essential for Alzheimer's treatment. BRAF inhibitor/RG6344 is not a P-glycoprotein substrate, potentially enhancing its bioavailability, while the other two drugs are substrates, possibly affecting their pharmacokinetics. Drugs 388 (BI-D1870) and 610 (Iloperidone) inhibit several CYP enzymes, which may lead to drug-drug interactions, while BRAF inhibitor/RG6344 does not, making it a safer option for metabolic interactions. All three drugs are Lipinski compliant, with having 0 violation, suggests that they all possess favorable pharmacological properties for oral administration. In addition to Lipinski's Rule of Five, which primarily assesses oral bioavailability, we employed more stringent filters to prioritize compounds with a low risk of promiscuous activity and

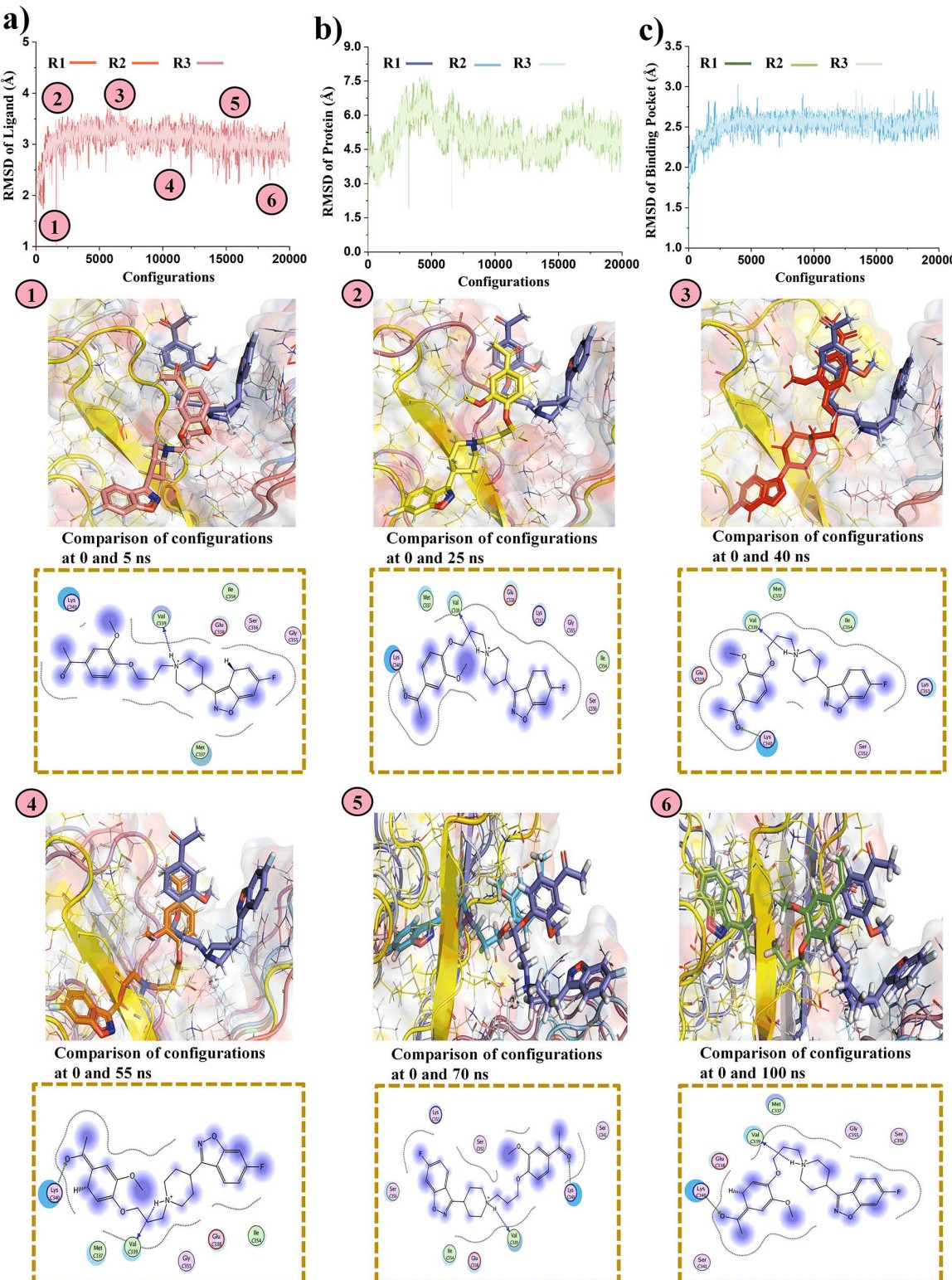

**Fig 6. Molecular dynamics simulation analysis of the Tau protein with the target Drug 610 (Iloperidone/HP 873) drug (F-18 labeled).** 2D interactions diagrams shows the hydrogen bonding and hydrophobic interactions between the drug and the receptor. **a)** the RMSD of drug (Drug 610 (Iloperidone/HP 873)), **b)** the RMSD of the receptor protein, and **c)** the RMSD of the binding region. The RMSD investigation validates the robust binding, which is crucial for the validation of the drug's stability within the Tau binding pocket.

**Table 2. SD, CV, and p-Values for RMSD Metrics Across Drugs** The table below summarizes the standard deviation (SD), coefficient of variation (CV = SD/mean × 100%), and p-values for the equilibrated phase (last 10% of simulations). Means and SDs are replica-pooled (n ≈ 660 per drug). ANOVA p-values are overall across drugs per metric (from one-way ANOVA). Tukey HSD p-values are adjusted pairwise (all < 0.001, indicating significance for every comparison: 318 vs. 416, 318 vs. 610, 416 vs. 610).

| RMSD Metric | Drug | Mean (Å) | SD (Å) | CV (%) | ANOVA p-value | Tukey HSD p-values (all pairs) |
|---|---|---|---|---|---|---|
| Ligand | 318 | 0.895 | 0.143 | 16.0 | < 0.0001 | < 0.001 |
| | 416 | 1.548 | 0.122 | 7.9 | | < 0.001 |
| | 610 | 2.970 | 0.153 | 5.2 | | < 0.001 |
| Pocket | 318 | 2.094 | 0.061 | 2.9 | < 0.0001 | < 0.001 |
| | 416 | 1.189 | 0.128 | 10.8 | | < 0.001 |
| | 610 | 2.582 | 0.088 | 3.4 | | < 0.001 |
| Alpha Carbon | 318 | 10.543 | 0.569 | 5.4 | < 0.0001 | < 0.001 |
| | 416 | 5.959 | 0.528 | 8.9 | | < 0.001 |
| | 610 | 5.060 | 0.464 | 9.2 | | < 0.001 |

favorable CNS multiparameter optimization (MPO) profiles. Specifically, we implemented Pan-Assay Interference Compounds (PAINS) filtering to eliminate molecules containing substructures known to cause false-positive results in biological assays. These interfering motifs, such as certain enones, catechols, or rhodanines, can react non-specifically with protein targets or exhibit assay-dependent fluorescence, leading to misleading conclusions about true efficacy. The absence of PAINS alerts (0 alerts) for all three candidate drugs significantly increases confidence in their specific, target-driven binding to Tau. This suggests that these drugs are more likely to yield accurate results in biological assays, which is essential for their development as safe and effective therapeutic candidates (Table 3).

**Table 3. Physicochemical and pharmacokinetic properties of target drugs.**

| ADMET Properties | Drug 388 (BI-D1870) | Drug 416 BRAF inhibitor/RG6344 | Drug 610 (Iloperidone/HP 873) |
|---|---|---|---|
| Formula | C24H16F3N5 | C20H17F2N5O4S | C24H27FN2O4 |
| Molecular weight | 428.42 g/mol | 459.45 g/mol | 425.48 g/mol |
| Num. heavy atoms | 32 | 32 | 31 |
| Fraction Csp3 | 0.08 | 0.25 | 0.42 |
| Molar Refractivity | 116.68 | 115.89 | 119.98 |
| TPSA | 77.05 (Å²) | 125.70 Å² | 64.80 (Å²) |
| LogP | 2.64 | 2.05 | 2.72 |
| Water solubility | 4.82 | 3.48 | 5.47 |
| GI absorption (%) | High | High | High |
| BBB permeant | Yes | Yes | Yes |
| P-glycoprotein substrate | Yes | NO | Yes |
| CYP1A2 Inhibitor | Yes | NO | NO |
| CYP2C19 Inhibitor | Yes | NO | Yes |
| CYP2C9 Inhibitor | Yes | Yes | Yes |
| CYP2D6 Inhibitor | Yes | NO | Yes |
| CYP3A4 Inhibitor | Yes | NO | Yes |
| PAINS | 0 alert | 0 alert | 0 alert |
| Lipinski | Yes; 0 violation | Yes; 0 violation | Yes; 0 violation |
| Synthetic accessibility | 4.14 | 3.89 | 3.55 |

According to the toxicity analysis, Drug 388 (BI-D1870) exhibits the lowest toxicity risk, with most toxicity endpoints, such as CYP enzyme interactions, neurotoxicity, and respiratory toxicity, predicted to be inactive, suggesting a minimal likelihood of toxicity. Drug 416 (BRAF inhibitor/RG6344) presents a more mixed toxicity profile, with endpoints like CYP2D6, CYP2C9 interactions, and respiratory toxicity marked as active, indicating a moderate risk of toxicity. The probability values align with this, indicating a moderate likelihood of toxicity, particularly in carcinogenicity and immunotoxicity. Drug 610 (Iloperidone/HP 873) shows the highest potential for toxicity, with active predictions for multiple endpoints, including CYP2D6, CYP2C9, CYP3A4 inhibition, neurotoxicity, mutagenicity, and cardiotoxicity, indicating a higher probability of toxicity ([Fig 7](link)).

The probability values confirm that this drug carries the highest risk, particularly in CYP enzyme inhibition, neurotoxicity, and cardiotoxicity. In summary, Drug 388 is the safest in terms of toxicity, while Drugs 416 and 610 show some risks, with Drug 610 being of particular concern due to its higher toxicity predictions. The ADMET analysis reveals that all three F-18 labeled drugs have favorable pharmacokinetic and physicochemical properties for targeting Alzheimer's disease. They exhibit good GI absorption, BBB permeability, and with no PAINS alerts, which is crucial for their application in the central nervous system. These findings are consistent with the goal of developing compounds that can effectively cross the blood-brain barrier and reach their target site in the brain. Further, the optimal LogP range (2.0–3.0), and full compliance with Lipinski's rule of 5 indicate strong drug-likeness and efficient brain penetration. The absence of PAINS alerts enhances confidence in their target-specific activity, minimizing the risk of nonspecific interactions. Among the three, BI-D1870 exhibited the most balanced ADMET and lowest toxicity profile, suggesting a safer and more stable candidate for Tau PET probe development. RG6344 showed moderate metabolic stability, while Iloperidone's higher CYP inhibition potential indicates a need for careful pharmacokinetic optimization. Overall, these findings highlight the translational potential of the identified compounds as radiolabeled imaging agents capable of effective brain targeting with acceptable safety margins [57].

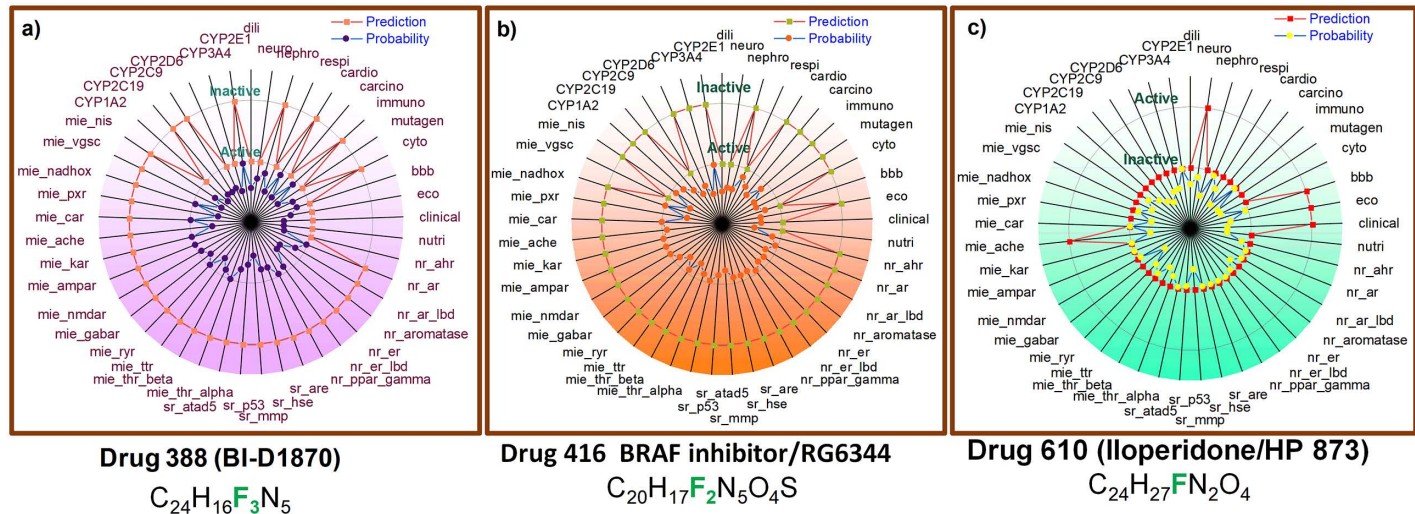

**Fig 7. Radar plots illustrating the predicted toxicity profiles for three drugs: (a) Drug 388 (BI-D1870), (b) Drug 416 (BRAF inhibitor/RG6344), and (c) Drug 610 (Iloperidone/HP 873).** The plots display toxicity predictions across multiple pharmacological endpoints, with active toxicity predictions highlighted in red and inactive predictions in purple. The probability of toxicity is represented by a color gradient from blue to green. These plots assess the potential risks of various adverse effects, including CYP enzyme inhibition, neurotoxicity, and cardiotoxicity.

## Comparative evaluation of the lead selected drugs for Tau in AD

Following computational investigations, binding associations, and ADMET evaluation, BI-D1870 (388) emerges as the most promising option for Tau-targeted imaging in AD with F-18 PET. BI-D1870 had the highest binding potential to Tau ($\Delta G = -8.79$ kcal/mol) and low RMSD oscillations in molecular dynamics simulations, indicating consistent and robust binding. Its effective ADMET characteristics, such as a LogP of 2.64, molecular weight of 428.42 g/mol, higher gastrointestinal absorption, high solubility (4.82), and robust blood-brain barrier (BBB) permeability, makes it an excellent option for PET imaging. Its F-18 labeling potential favors its diagnostics and therapeutic applicability in AD.

RG6344 (416), a BRAF inhibitors, has a significantly lower Tau binding affinity ($\Delta G = -7.91$ kcal/mol), yet continues to interacts strongly with important Tau residues. Its ADMET characteristics, including a LogP of 2.05, substantial gastrointestinal absorption, and high BBB permeability, make it a viable candidate for AD imaging. However, its significantly lower Tau binding affinity than BI-D1870 restricts its diagnostic usefulness. Its pharmacokinetic features are satisfactory, having a LogP value of 2.72, indicating adequate BBB penetration. Having a molecular weight of 425.48 g/mol, Iloperidone also follows the Lipinski Rule of Five. It has the greatest solubility of the three, at 5.47, which could aid in quicker absorption. Furthermore, it has significant GI absorption and BBB permeability. On the other hand, its lesser Tau binding affinity as in comparison with BI-D1870 and RG6344 limits its overall importance for Tau-targeted AD detection. In summary, BI-D1870 stands out as the most significant candidate for Tau-targeted imaging in AD due to its superior binding affinity with Tau receptor. Moreover, BI-D1870 is the most attractive option for PET imaging and possible therapeutic intervention in AD, despite the fact that RG6344 and Iloperidone exhibit promise due to their lower binding affinities and marginally less favorable ADMET profiles. The comparative evaluation clearly establishes BI-D1870 as the most promising Tau-targeted PET candidate, owing to its highest binding affinity, stable molecular dynamics profile, and optimal ADMET characteristics. The strong and consistent Tau interaction pattern of BI-D1870 suggests its potential for high target selectivity and imaging contrast, similar to benchmark tracers such as [$^{18}$F]PI-2620 and [$^{18}$F]MK-6240 [44]. RG6344 demonstrated moderate stability and affinity, indicating potential as a secondary lead, while Iloperidone's higher solubility but lower Tau binding limits its diagnostic efficiency. Collectively, these results confirm that BI-D1870 fulfills the key physicochemical and pharmacokinetic requirements for CNS PET tracers high binding affinity, low molecular weight, and optimal BBB permeability supporting its advancement for experimental validation in Alzheimer's imaging studies.

## Conclusion and future directions

The computational strategy employed in this study has successfully identified and optimized F-18 labeled drug candidates for Tau imaging and Alzheimer's diagnosis. The drugs BI-D1870 (388), RG6344 (416), and Iloperidone (610) have substantial Tau binding capacity with $\Delta G$ values of $-8.79$ kcal/mol, $-7.91$ kcal/mol, and $-6.88$ kcal/mol, accordingly, indicating their potential for Alzheimer's disease diagnostics. Notwithstanding their intriguing qualities, numerous restrictions remain, such as the absence of clinical trials for BI-D1870 and RG6344, limiting their immediate use in clinical applications. Additionally, although in-silico calculations and virtual screening of 977 CNS-penetrant medicines revealed potent Tau binders, the compounds' practical potency in imaging research and capacity to be transformed into therapeutic interventions have yet to be confirmed. Future approaches include additional in vivo confirmation of these F-18 labeled medicines, pharmacokinetic optimization, and clinical studies to evaluate their potential for improved Alzheimer's early identification and diagnosis. Furthermore, increased screening of various CNS-penetrant compounds, as well as further improvement of drug design computational methods, can help to identify more potent candidates. Finally, the fabrication of these novel PET imaging substances offers the possibility of revolutionizing Alzheimer's diagnosis and opening new paths for targeted therapy approaches.

## Supporting information

**S1 Fig. Characterization of the Initial Library of BBB-Penetrant Drugs for Tau-Targeted Virtual Screening and 18F Labeling Potential a) Clinical trial phase distribution across broad research areas, highlighting the prevalence of preclinical and early-phase drugs in cancer-related investigations.** b) Top 10 most frequent biological pathways targeted by the compounds, with notable enrichment in apoptosis, autophagy, and PI3K/Akt/mTOR signaling. c) Overall distribution of clinical phases among the drug candidates, showing a concentration in early (Phase 1–2) and investigational stages. d) Clinical phase distribution stratified by therapeutic research area, showing that oncology-dedicated compounds dominate advanced clinical phases. e) Boxplot of molecular weight (M. Wt) distributions across research areas, demonstrating consistency around the drug-like space (~300–500 Da), with cancer-focused compounds exhibiting slightly broader variability. f) Scatterplot correlating molecular weight with clinical phase, indicating no strong relationship but suggesting a tendency for lower-weight molecules to advance into later phases. g) Top 3 pathways associated with selected key drugs (SKF-96365, CG-806, CUDC-907, Cyclin-Dependent Kinase Inhibitors, and Semaxanib) show target diversity and relevance to neurodegeneration-related mechanisms.
(PDF)

**S2 Fig. The drug-tau protein complex interaction analysis.** a) 2D & 3D interaction diagram of the 2nd pose of the 388-drug within the active binding pocket of the Tau receptor. b) 2D & 3D interaction diagram of the 3rd pose of the 388-drug within the active binding pocket of the Tau protein. These diagrams offer in-depth perspectives of the molecular interactions, showing the important amino acids that contribute to the binding affinity as well as how the medication attaches to the tau protein.
(PDF)

**S3 Fig. The drug-tau protein complex interaction analysis.** a) 2D & 3D interaction diagram of the 2nd pose of the 416-drug within the active binding pocket of the Tau receptor. b) 2D & 3D interaction diagram of the 3rd pose of the 416-drug within the active binding pocket of the Tau protein. Evaluating the drug's potential for addressing tau-related neurodegenerative illnesses requires knowledge of the molecular underpinnings of the drug-tau interactions, including the potency of these interactions and important binding residues.
(PDF)

**S4 Fig. The drug-tau protein complex interaction analysis.** a) 2D & 3D interaction diagram of the 2nd pose of the 416-drug within the active binding pocket of the Tau receptor. b) 2D & 3D interaction diagram of the 3rd pose of the 416-drug within the active binding pocket of the Tau protein.
(PDF)

**S5 Fig. The energies profile associated with the interaction between the drug 388 and Tau protein is as follows: a) total energy (H), b) potential energy (U), and c) kinetic energy (K) in kcal/mol during a simulation of 100 ns.** The MD analysis indicates that the system reaches equilibrium and stays constant during the simulation since the total potential energy (H) and kinetic energy (K) are essentially constant.
(PDF)

**S6 Fig. During a 100 ns molecular dynamics simulation, the 416-Tau interaction's energy profile in kcal/mol is as follows: a) total energy (H), b) potential energy (U), and c) kinetic energy (K).** The MD study shows that the total internal energy (U) and the kinetic energy (K) are mostly stable, implying that the system achieves stability and remains constant throughout the experiment. However, the variations in potential energy (H) indicate that the system is adapting when 416 binds to Tau.
(PDF)

**S7 Fig. During a 100 ns molecular dynamics simulation, the 610-Tau interaction's energy profile is shown in kcal/mol as a) total energy (H), b) potential energy (U), and c) kinetic energy (K).** The MD evaluation shows that the total potential energy (H) and the kinetic energy (K) are virtually stable, implying that the system has reached equilibrium and will remain constant across the experiment. However, variations in internal energy (U) indicate that the system is adapting when CBD binds to Tau.
(PDF)

**S1 Table. Comparisons RMSD Ligand.** All pairs show significant differences, with RMSD increasing from drug 318 (tightest binding) to 610 (most flexible ligand).
(PDF)

**S2 Table. RMSD Pocket.** All pairs differ significantly, with drug 416 showing the most stable pocket, while 318 and 610 exhibit greater dynamics (610 > 318).
(PDF)

**S3 Table. RMSD of Protein.** All pairs are significantly different, with drug 318 showing the most protein backbone flexibility, followed by 416 and 610 (most rigid).
(PDF)

**S4 Table. Time-Averaged RMSD Values (Equilibrated Phase, Å) These confirm the trends: Drug 318 has the lowest ligand RMSD but highest Protein RMSD; drug 610 shows the opposite pattern.**
(PDF)

## Author contributions

**Conceptualization:** Bin Tang.

**Data curation:** Ling Zhang, Jun Wen.

**Formal analysis:** Pan Tang, Ling Zhang, Jun Wen, Iqra Kalsoom.

**Funding acquisition:** Chong Cheng.

**Investigation:** Bin Tang, Jun Wen.

**Methodology:** Pan Tang, Ling Zhang, Jun Wen, Iqra Kalsoom.

**Resources:** Chong Cheng.

**Software:** Xuehua Chen, Pingping Li, Yan Liu, Iqra Kalsoom.

**Supervision:** Iqra Kalsoom, Chong Cheng.

**Validation:** Pan Tang, Pingping Li, Bin Tang, Yan Liu.

**Visualization:** Xuehua Chen, Pingping Li, Yan Liu.

**Writing – original draft:** Xuehua Chen.

**Writing – review & editing:** Pan Tang, Ling Zhang, Iqra Kalsoom.

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
