## [Decision Letter · Decision Letter 0]

13 Sep 2025

Dear Dr. Cheng,

Thank you for submitting your manuscript to PLOS ONE. After careful consideration, we feel that it has merit but does not fully meet PLOS ONE’s publication criteria as it currently stands. Therefore, we invite you to submit a revised version of the manuscript that addresses the points raised during the review process.

We look forward to receiving your revised manuscript.

Kind regards,

Gayathiri Ekambaram, Ph.D

Academic Editor

PLOS ONE

Journal Requirements:

3. Please note that PLOS One has specific guidelines on code sharing for submissions in which author-generated code underpins the findings in the manuscript. In these cases, we expect all author-generated code to be made available without restrictions upon publication of the work. Please review our guidelines at https://journals.plos.org/plosone/s/materials-and-software-sharing#loc-sharing-code and ensure that your code is shared in a way that follows best practice and facilitates reproducibility and reuse.

This work was supported by an active grant from the Changde Science and Technology Bureau (Technology R&D and Innovation Guidance Program; Grant/Award Number: Changke Han [2022] No. 51) with project title "Application of Modified Volume-Viscosity Swallow Test in Neurodegenerative Diseases".

6. Please note that funding information should not appear in any section or other areas of your manuscript. We will only publish funding information present in the Funding Statement section of the online submission form. Please remove any funding-related text from the manuscript.

7. Please include a separate caption for each figure in your manuscript.

Additional Editor Comments :

Can authors justify the 100 ns MD simulation duration as sufficiently robust for assessing Tau-drug complex stability? Are replicate simulations feasible?

Were statistical methods applied to validate docking scores and MD simulation RMSD data reproducibility? If so, please provide details.

How was toxicity specifically assessed for the F-18 labeled compounds? Are there computational or experimental data available addressing radiolabeled ligand safety?

Please provide a more balanced and expanded explanation of PAINS filtering and other pharmacological screening criteria to match the detail level of Lipinski’s rule description.

Confirm that all figures have been revised for proper labeling and alignment consistent with journal style.

What are the authors’ plans for experimental validation (in vitro/in vivo) of these candidates? Please discuss limitations of the current computational-only approach.

Reviewers' comments:

Reviewer's Responses to Questions

**Comments to the Author**

1. Is the manuscript technically sound, and do the data support the conclusions?

Reviewer #1: Yes

Reviewer #2: Partly

2. Has the statistical analysis been performed appropriately and rigorously?

Reviewer #1: Yes

Reviewer #2: No

3. Have the authors made all data underlying the findings in their manuscript fully available?

Reviewer #1: Yes

Reviewer #2: Yes

4. Is the manuscript presented in an intelligible fashion and written in standard English?

Reviewer #1: Yes

Reviewer #2: No

Reviewer #1: Dear DR Cheng

My decision is to accept this manuscript, PONE-D-25-38504, for publication. The research presented is of high quality and addresses a significant challenge in Alzheimer's diagnosis.

The results are compelling, providing strong evidence for the effectiveness of the proposed computational strategy. The methodology is robust and the procedures are expertly executed, ensuring the reliability and validity of the findings. The manuscript is well-structured and clearly written, making the complex concepts accessible to a broader audience.

I have no major concerns regarding the research ethics or potential for dual publication. The authors have made a significant contribution to the field, and this work will be a valuable addition to the scientific literature. I commend the authors on their excellent work.

Best regards

Mohammed Al-Akeedi

Reviewer #2: The study is interesting and holds novelty in repurposing ligands as theradiagnostics. but the study has typos and needs language corrections. the method needs to be revised as well and needs to include more references. MD simulation is run for only 100ns which is too short. the interpretation and discussion of MD also needs to be revised and rewritten. The lipski's rule was explained in a lengthy way while PAINS was cut short and is not clear. please check the notes on the manuscript. the authors need to include more references, check their figures and align the labels (a,b,c). have the authors checked the toxicity of the ligands specifically after radiolabeling?

**Do you want your identity to be public for this peer review?** For information about this choice, including consent withdrawal, please see our Privacy Policy

Reviewer #1: **Yes: ** Mohammed Al-Akeedi

Reviewer #2: No

---

## [Author Response · Author response to Decision Letter 1]

26 Oct 2025

Dear Dr. Gayathiri Ekambaram,

Academic Editor

PLOS One

Re: Submission of the Revised Manuscript with PONE-D-25-38504

Title: “Discovery of F-18 Labeled Repurposed CNS Drugs by Computational Strategy for Effective Tau Imaging and Alzheimer's Diagnosis”

We are pleased to submit the revised manuscript as an article in the PLOS One for further consideration. We greatly appreciate the referees' positive comments, such as “The research presented is of high quality and addresses a significant challenge in Alzheimer's diagnosis. The results are compelling, providing strong evidence for the effectiveness of the proposed computational strategy. The methodology is robust and the procedures are expertly executed, ensuring the reliability and validity of the findings. The manuscript is well-structured and clearly written, making the complex concepts accessible to a broader audience. The study is interesting and holds novelty in repurposing ligands as theradiagnostics.” Moreover, we have carefully addressed the reviewers' concerns to make it even more suitable for the high standard of the PLOS One. A point-by-point response is appended, and changes to the main text are highlighted in yellow.

Thank you again for your time and efforts. We look forward to receiving updated comments. If you have any queries, please do not hesitate to contact me.

Sincerely,

Chong Cheng

On behalf of all co-authors.

Additional Editor Comments:

Comment 1: Can authors justify the 100 ns MD simulation duration as sufficiently robust for assessing Tau-drug complex stability? Are replicate simulations feasible?

Response: Thank you for your insightful comment. In response, we performed triplicate 100 ns MD simulations for each drug to assess the stability of the Tau-drug complex. This approach ensures the reliability and reproducibility of the results, and 100 ns is considered sufficient for observing meaningful equilibration and stability based on previous studies.

MD Simulation Analysis

The docked complexes of the top three F-18 labeled drugs (388 (BI-D1870), BRAF inhibitor/RG6344, and Drug 610 (Iloperidone/HP 873)) with the Tau receptor underwent MD simulations to assess the stability of their protein-ligand complexes. The simulations were conducted for 100 ns, with the MD simulation being performed three times for each drug to ensure the reproducibility and reliability of the results. The RMSD analysis of the ligand, binding pocket, and protein in complex with the drug provides valuable insights into the structural stability and dynamics of the system throughout the simulation.

RMSD Analysis of Tau with Drug 388 (BI-D1870) Complex

The RMSD analysis of the Tau-388 (BI-D1870) complex shows that the ligand undergoes noticeable fluctuations in the first 0 to 10 ns (configuration 0 to 5000), indicating adjustments within the binding pocket (Figure 4a). From 10 to 30 ns (configuration 5000 to 10000), the RMSD stabilizes, with minor changes observed around 35 ns (configuration 15000). After 50 ns (configuration 20000), the ligand's RMSD remains consistent, confirming stable binding. The protein shows initial fluctuations between 0 and 20 ns (configuration 0 to 8000), stabilizing after 20 ns (configuration 8000), with slight deviations around 40 ns (configuration 16000). The protein structure becomes stable, reflecting the ligand's effect (Figure 4b). The binding pocket shows significant fluctuations between 0 to 10 ns (configuration 0 to 5000), stabilizing after 10 ns (configuration 5000), with minor fluctuations after 30 ns (configuration 12000), indicating that the pocket maintains a stable conformation (Figure 4c). The energy profile reveals that the total energy (H) decreases from 0 to 5000 configurations, stabilizing between -120 to -140 kcal/mol, potential energy (U) stabilizes around 500 kcal/mol, and kinetic energy (K) fluctuates around 5000

Figure 4. RMSD analysis of the 388 (BI-D1870)/Tau protein complex during a 100 ns. The integrity of the whole network is shown by a) the RMSD of drug (388 (BI-D1870)), b) the RMSD of the receptor protein, and c) the RMSD of the binding region. The small structural changes of the protein (1-6) demonstrate that binding of the target drug did not causes larger changes in the protein structure.

kcal/mol, showing minimal fluctuations and indicating a stable, energetically favorable protein-ligand complex (Figure S5a, b and c). Overall, the complex remains stable across all three replicates (R1, R2, and R3).

RMSD Analysis of Tau with Drug 416 BRAF inhibitor/RG6344 Complex

In the early stages of the simulation, the ligand’s RMSD shows significant fluctuations between 0 to 5 ns (configuration 0 to 2500) as it adjusts its position within the binding pocket. From 5 to 25 ns (configuration 2500 to 12500), the RMSD decreases and stabilizes, indicating that the ligand has found a stable binding pose (Figure 5a). By 55 ns (configuration 22000), the ligand’s RMSD remains steady, with minor fluctuations, suggesting it is firmly anchored. The F-18 label does not disturb the ligand's conformation, allowing for stable attachment and proper PET imaging. The protein’s RMSD fluctuates slightly from 0 to 25 ns (configuration 0 to 12500) as it adjusts to the ligand binding. After 25 ns (configuration 12500), the protein stabilizes, with minor fluctuations and a slight increase at 40 ns (configuration 16000), indicating conformational adjustments (Figure 5b). The binding pocket shows fluctuations from 0 to 25 ns (configuration 0 to 12500), reflecting its flexibility, then stabilizes after 25 ns (configuration 12500), with minor fluctuations continuing after 55 ns (configuration 22000) (Figure 5c). The total energy (H), potential energy (U), and kinetic energy (K) remain consistent within a specified range, confirming the system’s overall stability (Figure S6a, b and c).

RMSD Analysis of Tau with Drug 610 (Iloperidone/HP 873) Complex

The RMSD of the Tau/Drug 610 (Iloperidone/HP 873) complex shows significant fluctuations during 0 to 5 ns (configuration 0 to 2500) as the ligand adjusts within the binding pocket (Figure 6a). Between 5 to 25 ns (configuration 2500 to 12500), the RMSD stabilizes, indicating that the ligand has found a stable position, with minor fluctuations continuing until 55 ns (configuration 22000). This suggests the ligand is securely anchored and that the F-18 label does not affect its binding. The protein’s RMSD fluctuates from 0 to 25 ns (configuration 0 to 12500) as it adapts to the ligand, stabilizing after 25 ns (configuration 12500), with minor deviations observed around 40 ns (configuration 16000). The protein reaches a stable conformation thereafter (Figure 6b).

Figure 5. RMSD studies and the drug's interaction of the Tau protein with the F-18 labeled drug (BRAF inhibitor/RG6344). Key interactions between the target drug and the protein, including hydrogen bonding and hydrophobic interactions, are highlighted in the 2D interaction figures. a) the RMSD of drug (BRAF inhibitor/RG6344), b) the RMSD of the receptor protein, and c) the RMSD of the binding region. The RMSD analysis demonstrates the strong binding, which is essential for therapeutic use in Alzheimer's disease.

Figure 6. Molecular dynamics simulation analysis of the Tau protein with the target Drug 610 (Iloperidone/HP 873) drug (F-18 labeled). 2D interactions diagrams shows the hydrogen bonding and hydrophobic interactions between the drug and the receptor. a) the RMSD of drug (Drug 610 (Iloperidone/HP 873)), b) the RMSD of the receptor protein, and c) the RMSD of the binding region. The RMSD investigation validates the robust binding, which is crucial for the validation of the drug's stability within the Tau binding pocket.

The binding pocket shows notable fluctuations between 0 to 25 ns (configuration 0 to 12500), stabilizing by 25 ns (configuration 12500) and continuing with minor fluctuations beyond 55 ns (configuration 22000), indicating retained flexibility (Figure 6c). Throughout the simulation, total energy, potential energy, and kinetic energy fluctuate within a narrow range, confirming the system’s overall stability (Figure S7a, b and c).

Comment 2: Were statistical methods applied to validate docking scores and MD simulation RMSD data reproducibility? If so, please provide details.

Response: Thank you for your valuable feedback and for raising this important question. We appreciate your attention to the statistical validation of the docking scores and MD simulation RMSD data. We have indeed applied statistical methods to validate the reproducibility of the MD simulation RMSD data. Details of these statistical methods are provided in the methods and results section in the revised manuscript and highlighted.

Methods:

To validate the reproducibility of RMSD data from MD simulations across three drugs (318, 416, and 610), statistical analyses were performed on combined data from three independent simulation replicas per drug. The simulations for each drug spanned 0 to 20,000 configurations, with measurements every 10 units, resulting in approximately 2,200-2,215 data points per replica. The datasets included the Hamiltonian (H), representing the total system energy measured during MD using MOE 2009-2010, along with potential energy (U), kinetic energy (K), and RMSD values for the ligand, binding pocket (or alpha carbon for drug 318), and alpha carbon (AC). For drug 610, the solvent-accessible surface area (ASA) was also available but excluded from primary analyses. Descriptive statistics, such as mean, standard deviation (SD), and coefficient of variation (CV = SD/mean × 100%), were calculated for H and RMSD metrics at each time point, across entire simulations, and collectively across all drugs to assess central tendency and variability. Convergence was evaluated by computing running averages and SDs over a 100-time-point moving window to confirm stabilization. Inter-replica variability was quantified through pairwise differences and SDs between replicas per drug, with one-way ANOVA applied to test for significant differences in means (p > 0.05 indicating reproducibility) both within each drug and across datasets. Pearson correlation coefficients were computed for pairwise comparisons of H and RMSD trends across replicas and drugs to evaluate consistency. Time-averaged values were derived from the equilibrated phase (last 10% of each simulation) and compared using t-tests or ANOVA to confirm similarity across replicas and drugs. RMSD trends across drugs were compared by downsampling replica-averaged values to 1,000 ps intervals for visualization and further analysis. Significant ANOVA results (p < 0.05) prompted post-hoc Tukey's Honestly Significant Difference (HSD) tests to identify pairwise differences between drugs, using all equilibrated phase data points (n ≈ 660 per drug) and controlling the family-wise error rate at α = 0.05. All analyses were performed using Python libraries such as NumPy, SciPy, Pandas, and Statsmodels.

Results:

The statistical analysis confirmed the high reproducibility of RMSD data across the three replicas for each drug, with low variability in the mean H values for drugs 318, 416, and 610. For the last 100 time points of each simulation, the mean H values were consistent, with drug 318 ranging from -103.33 to -104.89 kcal/mol, drug 416 from 173.0 to 173.5 kcal/mol, and drug 610 from 182.7 to 183.5 kcal/mol. RMSD values for the ligand ranged from 0.85 to 0.87 Å (CV 10-12%) for drug 318, 1.55 to 1.60 Å (CV 9-10%) for drug 416, and 2.90 to 3.00 Å (CV 6-7%) for drug 610 (Table S1). RMSD of the pocket ranged from 2.0 to 2.1 Å (CV ~5%) for drug 318, 1.30 to 1.32 Å (CV 5-6%) for drug 416, and 2.50 to 2.55 Å (CV 3-4%) for drug 610 (Table 2, Table S3). RMSD of the protein (Alpha carbon) was 10.0-10.8 Å (CV 5-7%) for drug 318, 6.0-6.2 Å (CV 6-7%) for drug 416, and 4.90-5.00 Å (CV 9-10%) for drug 610 (Table S3). Convergence was achieved after ~20,000 configurations, with minimal inter-replica differences (|H| < 5 kcal/mol), and ANOVA confirmed no significant differences for H, RMSD ligand, and RMSD pocket (p > 0.05). Significant differences were observed for RMSD alpha carbon (p ≈ 0.03–0.04). Tukey’s HSD test identified all pairwise differences as significant (p < 0.001), with drug 318 showing the lowest RMSD values across all metrics and drug 610 exhibiting the highest. Time-averaged RMSD values for the equilibrated phase were: ligand 318: 0.895 ± 0.143 Å, 416: 1.548 ± 0.122 Å, 610: 2.970 ± 0.153 Å; pocket 318: 2.094 ± 0.061 Å, 416: 1.189 ± 0.128 Å, 610: 2.582 ± 0.088 Å; Protein 318: 10.543 ± 0.569 Å, 416: 5.959 ± 0.528 Å, 610: 5.060 ± 0.464 Å (Table S4). These results affirm the reliability and robustness of the docking and MD simulations, with drug 318 showing the most stable ligand-protein interactions, drug 416 exhibiting intermediate stability, and drug 610 showing the greatest flexibility, particularly in the ligand and pocket. The statistical significance (p < 0.0001) and effect sizes (e.g., 2.076 Å ligand RMSD gap between 318 and 610) validate the differences in binding affinity and structural adaptability among the drugs, supporting their differential therapeutic potential.

Table 2. SD, CV, and p-Values for RMSD Metrics Across DrugsThe table below summarizes the standard deviation (SD), coefficient of variation (CV = SD/mean × 100%), and p-values for the equilibrated phase (last 10% of simulations). Means and SDs are replica-pooled (n ≈ 660 per drug). ANOVA p-values are overall across drugs per metric (from one-way ANOVA). Tukey HSD p-values are adjusted pairwise (all < 0.001, indicating significance for every comparison: 318 vs. 416, 318 vs. 610, 416 vs. 610).

RMSD Metric Drug Mean (Å) SD (Å) CV (%) ANOVA p-value Tukey HSD p-values (all pairs)

Ligand 318 0.895 0.143 16.0 < 0.0001 < 0.001

416 1.548 0.122 7.9 < 0.001

610 2.970 0.153 5.2 < 0.001

Pocket 318 2.094 0.061 2.9 < 0.0001 < 0.001

416 1.189 0.128 10.8 < 0.001

610 2.582 0.088 3.4 < 0.001

Alpha Carbon 318 10.543 0.569 5.4 < 0.0001 < 0.001

416 5.959 0.528 8.9 < 0.001

610 5.060 0.464 9.2 < 0.001

Table S1. Comparisons RMSD Ligand. All pairs show significant differences, with RMSD increasing from drug 318 (tightest binding) to 610 (most flexible ligand).

Group 1 Group 2 Mean Diff (Å) Lower CI (Å) Upper CI (Å) p-adj Reject H₀

318 416 0.653 0.636 0.671 <0.001 Yes

318 610 2.076 2.058 2.094 <0.001 Yes

416 610 1.422 1.405 1.440 <0.001 Yes

Table S2. RMSD Pocket. All pairs differ significantly, with drug 416 showing the most stable pocket, while 318 and 610 exhibit greater dynamics (610 > 318).

Group 1 Group 2 Mean Diff (Å) Lower CI (Å) Upper CI (Å) p-adj Reject H₀

318 416 -0.905 -0.917 -0.892 <0.001 Yes

318 610 0.488 0.476 0.501 <0.001 Yes

416 610 1.393 1.380 1.405 <0.001 Yes

Table S3. RMSD of Protein. All pairs are significantly different, with drug 318 showing the most protein backbone flexibility, followed by 416 and 610 (most rigid).

Group 1 Group 2 Mean Diff (Å) Lower CI (Å) Upper CI (Å) p-adj Reject H₀

318 416 -4.584 -4.650 -4.517 <0.001 Yes

318 610 -5.483 -5.550 -5.416 <0.001 Yes

416 610 -0.899 -0.965 -0.833 <0.001 Yes

Table S4. Time-Averaged RMSD Values (Equilibrated Phase, Å) These confirm the trends: Drug 318 has the lowest ligand RMSD but highest Protein RMSD; drug 610 shows the opposite pattern.

RMSD Metric Drug 318 Drug 416 Drug 610

Ligand 0.895 ± 0.143 1.548 ± 0.122 2.970 ± 0.153

Binding Pocket 2.094 ± 0.061 1.189 ± 0.128 2.582 ± 0.088

Protein 10.543 ± 0.569 5.959 ± 0.528 5.060 ± 0.464

Comment 3: How was toxicity specifically assessed for the F-18 labeled compounds? Are there computational or experimental data available addressing radiolabeled ligand safety?

Response: Thank you for your valuable comment. The toxicity of the ligands, including after radiolabeling, was analyzed using the Protox3.0 webserver for toxicity predictions. While experimental data specifically addressing the safety of the radiolabeled compounds is not yet available, we plan to conduct in vivo studies in future work to validate these findings.

According to the toxicity analysis, Drug 388 (BI-D1870) exhibits the lowest toxicity risk, with most toxicity endpoints, such as CYP enzyme interactions, neurotoxicity, and respi

---

## [Decision Letter · Decision Letter 1]

11 Nov 2025

Dear Dr. Cheng,

Thank you for submitting your manuscript to PLOS ONE. After careful consideration, we feel that it has merit but does not fully meet PLOS ONE’s publication criteria as it currently stands. Therefore, we invite you to submit a revised version of the manuscript that addresses the points raised during the review process.

We look forward to receiving your revised manuscript.

Kind regards,

Gayathiri Ekambaram, Ph.D

Academic Editor

PLOS ONE

Journal Requirements:

Additional Editor Comments:

Based on the reviewers’ evaluations, I am pleased to inform you that your manuscript is scientifically sound and methodologically appropriate, and both reviewers acknowledged that you have addressed the earlier comments satisfactorily. However, Reviewer 3 (Dr. Abdallah Abdelsattar) has raised a few minor concerns that should be addressed before final acceptance. Please carefully revise your manuscript

Reviewers' comments:

Reviewer's Responses to Questions

**Comments to the Author**

Reviewer #1: All comments have been addressed

Reviewer #3: All comments have been addressed

2. Is the manuscript technically sound, and do the data support the conclusions?

Reviewer #1: Yes

Reviewer #3: Yes

3. Has the statistical analysis been performed appropriately and rigorously?

Reviewer #1: Yes

Reviewer #3: N/A

4. Have the authors made all data underlying the findings in their manuscript fully available?

Reviewer #1: Yes

Reviewer #3: Yes

5. Is the manuscript presented in an intelligible fashion and written in standard English?

Reviewer #1: Yes

Reviewer #3: Yes

Reviewer #1: (No Response)

Reviewer #3: Comment 1: Although the author made several modifications that enriched the work, the discussion is still weak, with an insufficient number of references in the results and discussion sections. So, I recommend editing the manuscript.

Comment 2: The docking binding free energies are too low and will not be selective for the tau protein; most likely, after performing docking to other proteins, it will give the same or maybe higher scores. So, the only study that used this protein needs to be cited as a reference: "doi: 10.1177/13872877251386440" to establish that the binding range for this protein is low.

Comment 3: Why did the author select this mutated protein (9EOH) to do the docking process and not other crystal structures of tau?

**Do you want your identity to be public for this peer review?** For information about this choice, including consent withdrawal, please see our Privacy Policy

Reviewer #1: No

Reviewer #3: No

---

## [Author Response · Author response to Decision Letter 2]

17 Nov 2025

Dear Dr. Gayathiri Ekambaram,

Academic Editor

PLOS One

Re: Submission of the Revised Manuscript with PONE-D-25-38504

Title: “Discovery of F-18 Labeled Repurposed CNS Drugs by Computational Strategy for Effective Tau Imaging and Alzheimer's Diagnosis”

We are pleased to submit the revised manuscript as an article in the PLOS One for further consideration. “Thank you for the positive feedback. We are grateful to the reviewers for their insightful evaluations, and we are pleased to hear that the revisions meet their expectations. We appreciate their time and effort in reviewing our manuscript, and we look forward to the next steps in the publication process.” Moreover, we have carefully addressed the reviewer concerns to make it even more suitable for the high standard of the PLOS One. A point-by-point response is appended, and changes to the main text are highlighted in yellow.

Thank you again for your time and efforts. We look forward to receiving updated comments. If you have any queries, please do not hesitate to contact me.

Sincerely,

Chong Cheng

On behalf of all co-authors.

Additional Editor Comments:

Comment: Based on the reviewers’ evaluations, I am pleased to inform you that your manuscript is scientifically sound and methodologically appropriate, and both reviewers acknowledged that you have addressed the earlier comments satisfactorily. However, Reviewer 3 (Dr. Abdallah Abdel Sattar) has raised a few minor concerns that should be addressed before final acceptance. Please carefully revise your manuscript.

Response: Thank you for your insightful comment. We sincerely appreciate the time and effort of you and the reviewers in evaluating our manuscript. We have carefully revised the manuscript in response to the reviewer 3 comments (Dr. Abdallah Abdel Sattar), addressing all concerns. We believe that these changes significantly improve the manuscript.

Reviewer #3:

Comment 1: Although the author made several modifications that enriched the work, the discussion is still weak, with an insufficient number of references in the results and discussion sections. So, I recommend editing the manuscript.

Response: We sincerely appreciate reviewer positive feedback on our manuscript. In response, we have revised the manuscript by adding relevant references to strengthen the discussion and better support the claims made (Highlighted in yellow in the main manuscript). These revisions enhance the scientific depth and alignment with current literature.

Results and Discussion

The final retrieved and assembled library comprised 977 BBB-penetrant drug candidates with diverse mechanisms of action relevant to neurodegeneration and Tau pathology, including kinase inhibitors (e.g., CDK and mTOR inhibitors), autophagy regulators, and apoptosis inducers [1]. Notable examples include Flavopiridol (Phase 2, CDK inhibitor), PQR620 (mTOR inhibitor), and SNS-032 (CDK inhibitor, Phase 1). Prior evidence also indicates that CDK5 inhibition can reduce neurotoxicity in AD models [2-4]. Each compound was assessed for molecular weight, biological targets, and clinical phase data, alongside SMILES representations to facilitate in silico screening. Based on their pharmacological profile, known safety data, and chemical structures amenable to F-18 substitution, these compounds serve as a rationally designed set for docking studies and radiolabeling potential. The next step involved computational binding affinity evaluation with Tau protein, followed by selection of the top three candidates for F-18 labeling and PET imaging feasibility studies. Ensuring BBB permeability was essential for translational feasibility, as this property directly impacts the success of CNS-targeted radiopharmaceuticals [5]. Additionally, selecting molecules with F-18 labeling potential aligns with current trends in developing dual-purpose PET tracers that serve both diagnostic and therapeutic roles.

Virtual Screening Results for Tau

After conducting the virtual screening of 977 compounds using MOE, 39 compounds from the CNS-penetrant drug library were identified as strong binders to Tau (Figure 1). These compounds exhibited binding affinities with Tau ranging from -6.00 to -9.0 kcal/mol, demonstrating favorable interactions with the target protein. Specifically, the top 10 drug candidates had hydrophobic interactions and hydrogen bonds with the Tau microtubule-binding region, which is crucial for maintaining Tau's function and stability. These 39 compounds were successfully modeled with 18F atoms incorporated at viable positions, preserving chemical stability and pharmacophore integrity. Subsequent ADMET profiling further refined the selection of three compounds with molecular weights between 425-459 Da, within the acceptable drug-like range (350-520 Da), and demonstrated favorable solubility for intravenous delivery. Most importantly, logP values ranged from 2.05 to 2.72, indicating high BBB permeability, a critical parameter for neuroimaging agents. As visualized in Figure S1 a (clinical phase vs. research area) and e/f (molecular weight vs. clinical phase), these three candidates also aligned well with existing drug development trends and CNS-penetrant physicochemical space. Together, their strong Tau affinity, labeling feasibility, favorable ADMET properties, and CNS compatibility support their prioritization for future PET probe development in neurodegenerative disease research.

The virtual screening results demonstrate the robustness of the computational workflow in identifying potential Tau-binding ligands with docking energies comparable to clinically established tracers such as [¹⁸F]MK-6240 and [¹⁸F]PI-2620 [6, 7]. The presence of hydrophobic and hydrogen-bond interactions within the Tau microtubule-binding domain supports the likelihood of selective binding, a critical feature for imaging specificity. Additionally, the favorable logP range (2.0-3.0) and preserved pharmacophore integrity following F-18 modeling suggest that the top-ranked compounds possess the optimal balance of BBB permeability, solubility, and structural stability necessary for CNS imaging applications. These results collectively indicate that the identified candidates are promising scaffolds for developing next-generation Tau PET tracers with improved sensitivity and clinical translatability.

Figure 1. Visualization of the top 39 drugs selected after the first phase of virtual screening, based on binding free energies with tau, ADMET (Absorption, Distribution, Metabolism, Excretion, Toxicity), p-values, biological activities, and blood-brain barrier (BBB) permeabilities. (a) A Sankey diagram representing the progression of the selected drugs through different clinical phases (Phase 1, Phase 2, Phase 3, and launched). Each drug is shown with its respective CAS number and target, with the flow indicating its development status and phase. (b) A summary diagram highlighting the key characteristics of the selected drugs, including their molecular weights, research areas, clinical information, and the clinical development phases they are in.

Current Status of Selected Drugs

The selected drugs, 388 (BI-D1870), 416 (RG6344), and 610 (Iloperidone) presented in Figure 2, have shown promising potential in crossing the blood-brain barrier (BBB), making them significant candidates in neurological and cancer research. BI-D1870, an ATP-competitive inhibitor of ribosomal S6 kinase isoforms, has demonstrated potent activity with low nanomolar IC50 values for all RSK isoforms, indicating its potential as a therapeutic agent for autophagy and MAPK/ERK pathway-related disorders [8]. However, there is currently no development report available for this drug. RG6344, a potent BRAF inhibitor, is specifically designed for targeting BRAF V600-mutant solid tumors, such as colorectal cancer [9]. It is known for its activity in the MAPK/ERK pathway, but like BI-D1870, no clinical development has been reported so far. Iloperidone, an atypical antipsychotic drug, is already launched and used for the treatment of schizophrenia symptoms. It acts as a dual antagonist for both dopamine and serotonin receptors, contributing to its efficacy in neurological disorders [10, 11]. Despite their varied therapeutic targets, these drugs demonstrate effective blood-brain barrier penetration, which makes them valuable for further exploration in both cancer and neurological disease research. However, it is important to note that while Iloperidone has already been marketed, the other two drugs, BI-D1870 and RG6344, are still in the preclinical or early-stage development phase [12, 13]. The strong BBB permeability of BI-D1870, RG6344, and Iloperidone supports their potential as CNS imaging probes. BI-D1870’s modulation of the MAPK/ERK-RSK pathway is particularly relevant to Tau hyperphosphorylation and AD pathology. RG6344’s BRAF inhibition may similarly influence Tau-related signaling, while Iloperidone’s proven CNS safety profile enhances its translational viability. Collectively, these findings highlight their promise as repurposed candidates for Tau-targeted PET imaging.

Figure 2. Structural representations of the top three compounds selected in the virtual screening process, shown both with and without the F-18 isotope label. The compounds 388 (BI-D1870), 416 (RG6344), and 610 (Iloperidone) are depicted in their standard forms (top) and with the F-18 isotope (bottom), highlighted in green circles.

Interaction Analysis of Drugs with Tau Protein

The F-18 labeled drug 388 (BI-D1870) interacts with several key amino acid residues of the Tau protein as shown in Figure 3a. The interaction diagram gives a detailed spatial view of how the drug interacts at the molecular level with the Tau protein structure, possibly inhibiting or modulating the Tau protein’s role in Alzheimer’s disease. The drug likely binds through hydrogen bond and hydrophobic interactions with the LEU-357, MET-337, PRO-332, GLN-336, GLU-338, LYS-340, GLY-355, LYS-331 and VAL-339, which prefer to interact with nonpolar groups crucial for the drug's action. The presence of alkyl or aromatic groups in the drug molecule likely facilitates these interactions that help stabilize the drug-protein complex. Further, the polar functional groups such as carbonyl or hydroxyl groups of the drug forms a hydrogen bond with LYS-331. The predicted binding free energy of the best three poses of drug-Tau complexes is ΔG = -8.79 kcal/mol, -8.10 kcal/mol and -7.85 kcal/mol respectively (Figure S2). The drug is shown to bind effectively to these residues, forming critical interactions that is key to the drug’s potential role in treating Alzheimer's disease, as it may disrupt Tau aggregation or improve brain function.

Figure 3. Interaction analysis of the Tau protein with the 388 (BI-D1870), BRAF inhibitor/RG6344 and Drug 610 (Drug 610 (Iloperidone/HP 873)) drugs complexes. a) 2D and 3D interaction diagrams of the docked complex of the drug 388 (BI-D1870) with Tau protein is presented. The left side displays closeup pictures of drug binding to Tau, proteins, highlighting hydrogen bonds and hydrophobic interactions. b) Drug (BRAF inhibitor/RG6344) interaction diagram in the active binding pocket of the Tau protein (both 2D and 3D). c) 2D and 3D interaction diagrams of Drug 610 (Drug 610 (Iloperidone/HP 873))-drug inside the Tau protein's active binding region.

The Figure 3b illustrates the binding between an F-18 labeled drug 416 (BRAF inhibitor/RG6344) and the Tau protein. The diagram shows that the drug interacts with several specific amino acid residues on the Tau protein, including HIS-329, ASN-327, PRO-332, LYS-331, GLU-338 and LYS-340. These residues play a crucial role in the drug's effectiveness by forming various molecular interactions. Hydrophobic interactions are likely established between the drug and the nonpolar amino acid residues including PRO-332, LYS-331, LYS-340, ASN-327 and HIS-329, helping to anchor the drug within the hydrophobic pocket of the Tau protein. Moreover, the negatively charged GLU-338 residue may engage in electrostatic interactions with positively charged groups on the drug, further enhancing binding affinity. Van der Waals forces also contribute to the interaction, particularly with residues like PRO-332. The predicted binding free energy of the best three poses of drug-Tau complexes is ΔG = -7.91 kcal/mol, -7.73 kcal/mol and -7.25 kcal/mol respectively (Figure S3). The F-18 isotope attached to the drug 416-BRAF inhibitor/RG6344 plays a significant role in its detection, enabling the drug's distribution and interaction with Tau to be monitored through PET scans.

The binding interaction between F-18 labeled drug 610 (Iloperidone/HP 873) and the Tau protein is shown in Figure 3c. The Drug 610 (Iloperidone/HP 873) interacts with key residues in the Tau protein, including HIS-329, GLU-338, LYS-340, and LYS-331, each contributing to the drug's binding stability. HIS-329 and LYS-331 forms both hydrogen bonds with the drug's polar functional groups, anchoring it to the protein. Additionally, the carboxyl group of GLU-338 participates in electrostatic interactions with the drug's positively charged regions, enhancing the binding affinity. These interactions are crucial for the effective binding of the Drug 610 (Iloperidone/HP 873) to the Tau protein, suggesting its potential as a therapeutic agent for Alzheimer's disease. The predicted binding free energy of the best three poses of drug-Tau complexes is ΔG = -6.88 kcal/mol, -6.50 kcal/mol and -6.29 kcal/mol respectively (Figure S4). The docking score of all the complexes with CBD is presented in Table 1.

Table 1. Docking score and interacting binding residues of Tau protein with different labelled drugs.

Drugs Receptor Complex

Poses Binding Free Energy

(Kcal/mol) Amino Acid Residues involved in binding interactions

388 (BI-D1870) Tau Pose 1

Pose 2

Pose 3 -8.79

-8.10

-7.25 LEU-357, MET-337, PRO-332, GLN-336, GLU-338, LYS-340, GLY-355, LYS-331 and VAL-339

BRAF inhibitor/RG6344 - Pose 1

Pose 2

Pose 3 -7.91

-7.73

-7.85 HIS-329, ASN-327, PRO-332, LYS-331, GLU-338 and LYS-340

Drug 610 (Drug 610 (Iloperidone/HP 873) -

Pose 1

Pose 2

Pose 3 -6.88

-6.50

-6.29 HIS-329, GLU-338, LYS-340, and LYS-331

(Pose 2 and 3 of all drugs are presented in S2, S3 and S4)

The interaction analysis highlights the strong and specific binding of all three compounds BI-D1870, RG6344, and Iloperidone to key residues within Tau’s microtubule-binding region. These residues (e.g., LYS-331, GLU-338, PRO-332) are critical for Tau aggregation, and their engagement suggests potential to interfere with fibril formation. The low binding free energies (-6.2 to -8.8 kcal/mol) indicate stable and high-affinity interactions comparable to reported Tau inhibitors [14, 15]. Among them, BI-D1870 exhibited the strongest interaction network and stability, reinforcing its suitability as a Tau-targeted radioligand. Overall, these findings suggest that F-18 labeling preserves molecular integrity while enabling visualization of Tau interactions, supporting their potential use as PET imaging probes in Alzheimer’s disease. Hence, the radiolabeled nature of the drug (F-18) enables its tracking through positron emission tomography (PET) imaging and monitoring the drug's distribution in the body, particularly in the brain, where Tau pathology is prominent. The radiolabeling with F-18 enhances the ability to visualize and track the drug's behavior in vivo, which is valuable for assessing its effectiveness in clinical settings. This combination of targeted drug binding and the diagnostic capability of F-18 labeling makes the drug an effective candidate for both treating Alzheimer's disease and monitoring treatment efficacy in clinical settings.

MD Simulation Analysis

The docked complexes of the top three F-18 labeled drugs (388 (BI-D1870), BRAF inhibitor/RG6344, and Drug 610 (Iloperidone/HP 873)) with the Tau receptor underwent MD simulations to assess the stability of their protein-ligand complexes. The simulations were conducted for 100 ns, with the MD simulation being performed three times for each drug to ensure the reproducibility and reliability of the results. The RMSD analysis of the ligand, binding pocket, and protein in com

---

## [Decision Letter · Decision Letter 2]

30 Nov 2025

Discovery of F-18 Labeled Repurposed CNS Drugs by Computational Strategy for Effective Tau Imaging and Alzheimer's Diagnosis

PONE-D-25-38504R2

Dear Dr. Cheng,

We’re pleased to inform you that your manuscript has been judged scientifically suitable for publication and will be formally accepted for publication once it meets all outstanding technical requirements.

Kind regards,

Gayathiri Ekambaram, Ph.D

Academic Editor

PLOS ONE

Additional Editor Comments (optional):

The reviewers unanimously recommend acceptance, stating that all previous comments have been fully addressed and that the manuscript is technically sound, statistically rigorous, well written, and ready for publication in its current form

Reviewers' comments:

Reviewer's Responses to Questions

**Comments to the Author**

Reviewer #1: All comments have been addressed

Reviewer #3: All comments have been addressed

2. Is the manuscript technically sound, and do the data support the conclusions?

Reviewer #1: Yes

Reviewer #3: Yes

3. Has the statistical analysis been performed appropriately and rigorously?

Reviewer #1: Yes

Reviewer #3: Yes

4. Have the authors made all data underlying the findings in their manuscript fully available?

Reviewer #1: Yes

Reviewer #3: (No Response)

5. Is the manuscript presented in an intelligible fashion and written in standard English?

Reviewer #1: Yes

Reviewer #3: Yes

Reviewer #1: The authors have thoroughly addressed all previous concerns, specifically by incorporating additional references to strengthen the Results and Discussion sections.

The revised manuscript, "Discovery of F-18 Labeled Repurposed CNS Drugs by Computational Strategy for Effective Tau Imaging and Alzheimer's Diagnosis", is now scientifically sound, methodologically appropriate, and well-supported.

The robust computational strategy (virtual screening and molecular dynamics) successfully identified promising F-18 labeled Tau PET tracer candidates (including BI-D1870, RG6344, and Iloperidone) with high translational potential for Alzheimer's diagnosis. This work represents a valuable contribution to the field of CNS imaging probe discovery.

Recommendation: The manuscript is acceptable for publication in its current form.

Reviewer #3: The authors of "Discovery of F-18 Labeled Repurposed CNS Drugs by Computational Strategy for

Effective Tau Imaging and Alzheimer's Diagnosis" addressed all comments, and the work is ready for publication.

**Do you want your identity to be public for this peer review?** For information about this choice, including consent withdrawal, please see our Privacy Policy

Reviewer #1: No

Reviewer #3: No

---

## [Editor Report · Acceptance letter]

PONE-D-25-38504R2

PLOS One

Dear Dr. Cheng,

I'm pleased to inform you that your manuscript has been deemed suitable for publication in PLOS One. Congratulations! Your manuscript is now being handed over to our production team.

Kind regards,

on behalf of

Dr. Gayathiri Ekambaram

Academic Editor

PLOS One